The effect of decay and lexical uncertainty on processing long-distance dependencies in reading

http://orcid.org/0000-0002-2180-9736 Stone Kate 1 stone@uni-potsdam.de
http://orcid.org/0000-0001-5925-5145 von der Malsburg Titus 1 2
http://orcid.org/0000-0002-6014-3260 Vasishth Shravan 1
1 Department of Linguistics, Universität Potsdam , Potsdam , Germany
2 Department of Brain and Cognitive Sciences, Massachusetts Institute of Technology , Cambridge, MA , USA
McArthur Genevieve
Electronic publication date: 2020 Dec 17
Publication date: 2020
Volume: 8
Electronic Location ID: e10438
Received 2019 Nov 27; Accepted 2020 Nov 6
Copyright: © 2020 Stone et al.
Copyright year: 2020
Copyright holder: Stone et al.
License: This is an open access article distributed under the terms of the Creative Commons Attribution License, which permits unrestricted use, distribution, reproduction and adaptation in any medium and for any purpose provided that it is properly attributed. For attribution, the original author(s), title, publication source (PeerJ) and either DOI or URL of the article must be cited.
License URL: https://creativecommons.org/licenses/by/4.0/

Keywords: Reading, Comprehension, Temporal decay, Preactivation, Long distance dependencies, Entropy, Psycholinguistics, Locality, Antilocality

Funding: University of Potsdam Graduate School This work was supported by a scholarship from the University of Potsdam Graduate School. The funders had no role in study design, data collection and analysis, decision to publish, or preparation of the manuscript.

==============================
To make sense of a sentence, a reader must keep track of dependent relationships between words, such as between a verb and its particle (e.g. turn the music down). In languages such as German, verb-particle dependencies often span long distances, with the particle only appearing at the end of the clause. This means that it may be necessary to process a large amount of intervening sentence material before the full verb of the sentence is known. To facilitate processing, previous studies have shown that readers can preactivate the lexical information of neighbouring upcoming words, but less is known about whether such preactivation can be sustained over longer distances. We asked the question, do readers preactivate lexical information about long-distance verb particles? In one self-paced reading and one eye tracking experiment, we delayed the appearance of an obligatory verb particle that varied only in the predictability of its lexical identity. We additionally manipulated the length of the delay in order to test two contrasting accounts of dependency processing: that increased distance between dependent elements may sharpen expectation of the distant word and facilitate its processing (an antilocality effect), or that it may slow processing via temporal activation decay (a locality effect). We isolated decay by delaying the particle with a neutral noun modifier containing no information about the identity of the upcoming particle, and no known sources of interference or working memory load. Under the assumption that readers would preactivate the lexical representations of plausible verb particles, we hypothesised that a smaller number of plausible particles would lead to stronger preactivation of each particle, and thus higher predictability of the target. This in turn should have made predictable target particles more resistant to the effects of decay than less predictable target particles. The eye tracking experiment provided evidence that higher predictability did facilitate reading times, but found evidence against any effect of decay or its interaction with predictability. The self-paced reading study provided evidence against any effect of predictability or temporal decay, or their interaction. In sum, we provide evidence from eye movements that readers preactivate long-distance lexical content and that adding neutral sentence information does not induce detectable decay of this activation. The findings are consistent with accounts suggesting that delaying dependency resolution may only affect processing if the intervening information either confirms expectations or adds to working memory load, and that temporal activation decay alone may not be a major predictor of processing time.

Introduction

Keeping track of dependent relationships between words in a sentence is a crucial step in understanding meaning. For example to understand the full meaning of a particle verb such as turn down, a reader must recognise that these two words form a dependency, even when they are separated by other sentence material, such as in turn the music down. One question is whether readers anticipate the lexical content of such dependencies, or whether they wait to construct meaning retrospectively once the identity of the second word is known. In particle verb constructions in particular, anticipating the lexical identity of the particle would be advantageous to interpreting a potentially large amount of intervening sentence material, which might otherwise be difficult without access to the full verb. The intervening material may itself further sharpen expectation about the identity of the particle (Levy, 2008; Hale, 2001), or may instead create additional working memory load and activation decay that negatively impacts processing (Van Dyke & Lewis, 2003; Ferreira & Henderson, 1991; Gibson, 1998; Lewis & Vasishth, 2005; Vasishth & Lewis, 2006). In this paper, we examine whether readers anticipatorily pre-activate the lexical context of verb-particle dependencies in German and how intervening material impacts this pre-activation. Specifically, since previous work on dependency processing has focused on working memory load and interference, we attempt to isolate the effects of activation decay.

Lexical pre-activation in long-distance dependency formation

Contextual cues in a sentence are used to predictively pre-activate probable words and features in memory, such that processing of a predictable word can begin before that word is seen (Kuperberg & Jaeger, 2016; DeLong, Urbach & Kutas, 2005; Van Berkum et al., 2005; Wicha, Moreno & Kutas, 2004; Nicenboim, Vasishth & Rösler, 2020). Pre-activation therefore represents a processing advantage at predictable versus unpredictable words, as reflected by shorter reading times (Ehrlich & Rayner, 1981; Staub, 2015; Kliegl et al., 2004) and decreased event-related potential (ERP) components (Kutas & Hillyard, 1980, 1984; Kutas & Federmeier, 2011). It has also been proposed that strong pre-activation may trigger pre-integration of a specific lexical item into the building sentence representation in working memory (Ness & Meltzer-Asscher, 2018; Lewis & Vasishth, 2005; Vasishth & Lewis, 2006).

However, evidence for the pre-activation of lexical content in long-distance dependency formation is sparse. While there is evidence that specific lexical items are pre-activated by their context, pre-activation in such studies is generally only tested at the immediately preceding word or within the noun phrase (DeLong, Urbach & Kutas, 2005; Van Berkum et al., 2005; Wicha, Moreno & Kutas, 2004; Nicenboim, Vasishth & Rösler, 2020). To investigate longer distance dependency formation, researchers have demonstrated evidence that the left anterior negative (LAN) ERP component is larger at the initiation of long versus short syntactic wh-dependencies, suggesting that anticipation of a long dependency leads to greater working memory load (Fiebach, Schlesewsky & Friederici, 2002; Phillips, Kazanina & Abada, 2005). Applied to lexical pre-activation, a study of Dutch particle verbs hypothesised that verbs that take a large number of possible particles (e.g. “spannen”, to tense, which can take at least seven particles) should trigger pre-activation of those particles, placing a larger demand on working memory than verbs with a small set size (e.g. “kleuren”, to colour, which can take only two; Piai et al., 2013). When a verb-particle dependency is initiated by a verb that takes particles, the LAN should therefore be larger for large- versus small-set verbs. Instead, the authors observed that while the LAN was larger for verbs that took particles than those that did not, it did not differ between small and large set size. The authors concluded that the particles themselves were not pre-activated, but rather that readers anticipated the possibility of a downstream particle and stored the verb to facilitate its retrieval if a particle was encountered. Another plausible interpretation is that readers anticipated a particle and generated a syntactic prediction for its position, but not for its specific lexical identity. Together, this evidence suggests that readers pre-activate the syntactic structure of long-distance dependencies, but not long-distance lexical content.

Reading time studies have offered a different perspective on long-distance lexical pre-activation: complex predicate constructions in Hindi and Persian succeeded in eliciting a set size-type difference in reading times, which were faster at a target verb when a specific verb continuation was predictable than when no specific verb was predictable (Husain, Vasishth & Srinivasan, 2014; Safavi, Husain & Vasishth, 2016). Although these studies measured reading times at the target verb, the sentence stimuli in the Hindi study—including the target verb—were identical across conditions. Only the head noun differed, meaning that reading time differences at the target verb could reasonably be attributed to differences in pre-activation at the noun, rather than to differences in integrating the verb into different contexts. There is thus evidence that readers may pre-activate the lexical content of particle verb-type dependencies, although findings are inconsistent.

Delaying dependency resolution

Dependencies in English tend to be resolved relatively quickly (Futrell, Mahowald & Gibson, 2015), but this is often not the case in languages such as Dutch, Hindi, Persian, and German. This means that if dependent lexical content is pre-activated, pre-activation must be sustained over a potentially large amount of intervening sentence material. Processing of the intervening sentence material can have either a facilitatory or a hindering effect on processing of the dependency, as proposed by different theoretical accounts.

A hindering effect of delaying dependency resolution is predicted by accounts suggesting that processing intervening sentence material places a larger demand on working memory. The introduction of new discourse referents in particular has been associated with a locality effect in dependency processing, where reading of the distant word becomes slower the more new discourse referents are introduced. Slowed reading is proposed to reflect the cost of storing and integrating the new referents (Gibson, 1998, 2000), retrieval interference (Lewis & Vasishth, 2005; Vasishth & Lewis, 2006), and/or decay of constituent activation over time (Gibson, 1998, 2000; Lewis & Vasishth, 2005; Vasishth & Lewis, 2006; Vosse & Kempen, 2000), all contributing to longer retrieval time at the distant word.

A facilitatory effect of delaying dependency resolution may occur when the additional sentence material provides additional information as to the position and the identity of the distant word. This results in easier processing of the distant word, as reflected in faster reading times, otherwise known as an antilocality effect (Vasishth & Lewis, 2006). The facilitatory effect of increasing distance is captured by surprisal theory. Surprisal theory provides an information theoretic account of the difficulty of processing each new word in a sentence, represented by the negative log probability of that word appearing given the preceding context (Levy, 2008; Hale, 2001). According to surprisal theory, the building context of a sentence generates a set of licensed continuations. Each new word encountered triggers an update of the probability distribution of these continuations, and the degree of update is proportional to the difficulty of processing the new word; that is the greater the update, the greater the processing difficulty or ‘surprisal’. In broader terms, this means the more constraining a sentence is, the fewer likely possible continuations it will have, meaning lower surprisal and easier processing at a predictable word. Conversely, at an unpredictable word, surprisal and thus processing difficulty will be higher. Thus, surprisal theory predicts that the greater the amount of information separating two dependent words, the more predictable and easy to process the distant word will become.

The sources underlying antilocality and locality effects—predictability and working memory load respectively—may even interact. There is some evidence that the negative effect of high working memory load may only be apparent in weakly predictive contexts and that otherwise, antilocality effects are observed (Husain, Vasishth & Srinivasan, 2014; Konieczny, 2000; Levy & Keller, 2013). For example in German, it was found that reading times at the clause-final verb of a relative clause were faster when the verb was delayed by one additional constituent than when it was not delayed (an antilocality effect), but that reading times slowed down when the verb was delayed by two additional constituents (a locality effect; Levy & Keller, 2013). The authors reasoned that the relative infrequency of adding the second constituent (according to a corpus analysis) actually reduced predictability, making the effects of increased working memory load more pronounced. Casting doubt on these results, however, is a replication attempt finding only locality effects, regardless of what information preceded the verb (Vasishth et al., 2018).

More direct tests of an interaction between predictability and working memory load have been conducted in Hindi and Persian. In Hindi, increasing the separation within noun-verb complex predicates facilitated the reading of highly predictable verbs, but slowed the reading of low-predictable verbs, suggesting that high predictability outweighed the effect of additional working memory load introduced by the intervening sentence material (Husain, Vasishth & Srinivasan, 2014). However, this load/predictability interaction was not replicated in analogous constructions in Persian, where higher working memory load induced by additional sentence material slowed reading of the distant verb, regardless of the verb’s predictability (Safavi, Husain & Vasishth, 2016). One difference between the Hindi and Persian studies was the type of information used to manipulate the separation distance of the complex predicate dependencies. The Persian study used a relative clause and a prepositional phrase as an intervener (Safavi, Husain & Vasishth, 2016). Both relative clauses and prepositional phrases introduce new discourse referents and interference, both of which are predicted to burden working memory resources and slow reading (Gibson, 1998, 2000; Lewis & Vasishth, 2005), although new discourse referents may not be the only source of slowing in longer dependencies (Gibson & Wu, 2013). In comparison, the separation in the Hindi experiments was increased with adverbials, which instead may have increased evidence for the position and lexical identity of the upcoming verb (Hale, 2001; Levy, 2008). Altogether, these findings suggest that while readers may pre-activate the lexical entry of an upcoming dependent word, if appearance of that word is delayed, its predictability may play an important role in how the intervening information impacts processing.

Temporal activation decay

The effects of increased working memory load via new discourse referents and retrieval interference on dependency processing are well known, but the effects of temporal activation decay are less well-studied. Decay is proposed to affect sentence processing in the following way: At any new word in a sentence, there may be a number of ways the sentence structure could plausibly continue. For example the sentence The secretary forgot… could continue with a direct object noun phrase (e.g. the files) or with a clause (e.g. that the student…). It has been proposed that both of these structures are activated, but that only one is pursued by the parser while the other is left to decay (Van Dyke & Lewis, 2003). Thus, if the parser pursues the sentence structure assuming an upcoming noun phrase, but instead encounters the word that…, the decayed structure must be reactivated and reading time at the word that will be slower than if the expected noun phrase had been encountered (Ferreira & Henderson, 1991; Gibson, 1998; Van Dyke & Lewis, 2003). In sentences where multiple structures are left to decay, the differing activation levels of these decayed constituents will play a role in determining how fast they can be reactivated. Even if the correct constituent is pre-integrated initially, its activation will also decay over time due to the finite amount of activation available to the parser (Lewis & Vasishth, 2005; Vosse & Kempen, 2000; Gibson, 1998, 2000).

The above example concerns plausible structural continuations of the sentence, but plausible continuations may also include the pre-activation of specific lexical items. For example in 1a below, the verb turn may trigger pre-activation of plausible sentence continuations, including a large number of frequent particles (turn off, turn on, turn around, turn over, etc.). If the sentence continues with the music, pre-activation should be constrained to a smaller group of plausible particles:

(1) a. Turn the music… [on, off, up, down]

b. Calm the situation… [down]

A specific particle may even be pre-integrated while the others are left to decay. If future input indicates that the wrong particle was pre-integrated, for example up instead of down, then down must be reactivated in order to repair the sentence, resulting in longer reading times at the particle. As the number of plausible lexical items increases, reading times should therefore become slower on average, because the probability that the parser pursues a parse with the wrong lexical item increases and reactivation of decayed items will be needed more often. Alternatively, the starting activation of down in 1a may be lower than that of down in 1b, because the latter context points strongly to down as the only plausible continuation. The stronger starting activation of down in 1b should mean that even as activation decays over time, it will still have stronger activation at matched points in the sentence than in 1a. Thus, overall, more predictable lexical items should be more resistant to the effects of decay than less predictable items.

While activation decay may be a factor in sentence processing, there is evidence to suggest that it is not a useful predictor of processing difficulty (Van Dyke & Johns, 2012; Engelmann, Jäger & Vasishth, 2019; Vasishth et al., 2019), and that longer word recall times and reduced accuracy over time are better explained by interference than decay (Lewandowsky, Oberauer & Brown, 2009). On the other hand, much of this evidence comes from computational modelling based largely on data from experiments testing interference rather than specifically testing decay. There are few empirical experiments specifically testing decay in isolation, even though it is generally assumed to affect word processing times in long-distance dependencies (Xiang et al., 2014; Ness & Meltzer-Asscher, 2019; Chow & Zhou, 2019). One empirical study demonstrated the effects of decay over and above those of interference (Van Dyke & Lewis, 2003), although the authors later attributed these results to interference (Van Dyke & Johns, 2012). Nonetheless, a basic account of temporal activation decay would predict that the longer the distance between two dependent words in a sentence, the greater the activation decay and processing difficulty. Furthermore, decay and processing difficulty should be most pronounced when predictability of the distant word is low. This contrasts directly with the surprisal account, which predicts that the further away the dependent word, the easier processing should become.

The current experiments

We tested the decay/predictability interaction using German particle verbs, which are complex predicates similar to the constructions used in previous studies of Hindi and Persian (Husain, Vasishth & Srinivasan, 2014; Safavi, Husain & Vasishth, 2016). German particle verbs are comparable to English particle verbs in that they are composed of a base verb (e.g. “räumen”, to tidy) and a particle (e.g. “auf”, up) which can be separated (Müller, 2002). In German, however, the particle must appear after the direct object if the verb is transitive, usually at the right clause boundary (e.g. “Er räumte den Raum auf” he tidied the room up, but not “*Er räumte auf den Raum” he tidied up the room; Müller, 2002). Particle verbs form a very strong dependency because the full meaning of the verb “aufräumen” (to tidy up) can only be interpreted once both the verb and particle are known. Delaying appearance of the particle therefore creates a very strong structural expectation if the context makes a particle necessary, but potentially also a strong lexical expectation for a specific particle. In English particle verb constructions, the delay between a base verb and its particle is usually not very long; consider to tidy up vs ?/* to tidy the mess left after the party on Saturday up. In German, however, long-distance separations are common.

To manipulate lexical predictability of the distant particle, we compared base verbs that could take a large number of particles (10+) with verbs that can take only a small number of particles (six or fewer). We hypothesised that the set of potential particles would be pre-activated at the verb and that a larger set of particles would create more uncertainty (weaker predictability) about the eventual identity of the particle. Large set verbs therefore formed a low predictability condition and small set verbs a high predictability condition. Note that throughout the remainder of the article, we use set size as a proxy for predictability. Set size also relates to entropy, which we introduce in detail as it becomes relevant in the Cloze Test section. To induce decay between the verb and its particle, we manipulated distance with a neutral adjectival modifier. Critically, the modifier added no interference or working memory load through the introduction of new discourse referents (Gibson, 1998, 2000; Lewis & Vasishth, 2005), and did not provide semantic clues about the lexical identity of the dependency resolution. Any effects of the intervener on reading time were therefore attributable to temporal decay alone.

The design was based on the study of Dutch particle verbs (Piai et al., 2013). The Dutch study found no evidence of a modulation of LAN amplitude according to set size. We reasoned, however, that the distinction between small and large particle set sizes may have been too small: small set verbs took two to three particles and large set verbs, at least five. We therefore categorised our German verbs into small set verbs that took up to six particles, and large set verbs that took at least 10 particles. The current experiments therefore tested the hypotheses that (i) verbs that take particles trigger pre-activation of those particles; (ii) that delaying the appearance of the particle would slow reading times through temporal decay; but that (iii) higher predictability would make reading times at the particle less likely to be affected by decay.

We tested the hypotheses in self-paced reading and eye tracking experiments, both to confirm that any effects seen were not limited to a particular experimental method, but also because the two methods provide complementary information. Self-paced reading has the advantage of forcing readers to view each word in the sentence, whereas eye tracking allows words to be skipped and re-read. In the current study, the target word, a particle, was very short and may therefore have been more likely to be skipped, making self-paced reading data valuable in examining reading time effects at the particle. On the other hand, eye tracking has the advantage of more closely resembling natural reading and is able to measure phenomena such as regressive eye movements to previous regions of the sentence, and forward saccades to upcoming regions of the sentence. This allows us to generate hypotheses about the cognitive processes underlying slower or faster reading of a particular word and complements observations made in self-paced reading.

Predictions

It is well-established that more predictable words are associated with faster reading times than less predictable words, and thus we expected to see faster reading times for small set (more predictable) versus large set (less predictable) particles. With respect to our two distance conditions (short vs long), at short distance the predictions of surprisal theory and decay are the same: small set (more predictable) particles should be read faster than large set (less predictable) particles. This is reflected in both panels of Fig. 1, where predicted reading times for small set particles are always faster than those for large set particles.

Figure 1 Predicted interaction of lexical predictability (set size) and distance.

(A) Informal predictions of the surprisal account suggest that reading times will be faster for more predictable particles in the small set condition than less predictable particles in the large set condition. Reading times should always be faster at long distance due to increased expectation for the particle. (B) Predictions based on a simulation using the decay parameter of the LV05 model also suggest that reading times should be faster for more predictable particles in the small set condition. However, an effect of long distance should only be visible when predictability is low (large set), where activation decay should result in slower reading times at long versus short distance.

Where the predictions of surprisal theory and decay diverge is in the long-distance condition. Under surprisal theory, the long-distance condition should produce an antilocality effect (faster reading times) at both small set and large set particles, as illustrated in Fig. 1A. We attempted to quantify these predictions by computing surprisal values for the particles; however, the particular particle verbs used in the experiment were likely too infrequent in the corpora used and the parser’s surprisal estimates were unreliable.1 Instead, Fig. 1A represents informal predictions for the surprisal account. In the absence of formal quantifications for whether surprisal theory would predict an antilocality effect for our sentences, these predictions should be taken as an approximation of surprisal theory’s general claim that long distance should always result in faster reading times and that higher lexical predictability should sharpen expectations (Levy, 2008).

In contrast, under a decay account, reading times in the long-distance conditions should depend on how predictable the particle is. For more predictable (small set particles), pre-activation should be stronger to begin with and thus less affected by decay at long distance, whereas weaker pre-activation for less predictable (large set) particles may be more susceptible to decay, resulting in a locality effect (slower reading times) at long versus short distance. To quantify the effect of decay on reading time, we conducted a simulation using the decay parameter of the LV05 model (Lewis & Vasishth, 2005). Note that the full LV05 model was not used as it is primarily a model of interference, which we were not testing in the current study. To quantify predictability in the simulation, we assumed a finite pool of spreading activation for all of the plausible particle continuations. Dividing the finite pool of spreading activation among fewer particles meant a higher starting activation per particle in the small set than in the large set condition. Figure 1B shows that the simulation predicted a larger slow-down between small and large set size in the long distance condition than in the short distance condition. Code for the simulation is included in the R script in the paper’s OSF repository, see “Appendix 1”.

Experiment 1: Self-Paced Reading

Methods

Participants

Experiment 1 included a total of 60 participants (14 male, mean age = 24 years, SD = 6 years, range = 18–55 years) recruited via an in-house database. Participants were screened for acquired or developmental reading or language production disorders, neurological or psychological disorders, hearing disorders, and visual limitations that would prevent them from adequately reading sentences from the presentation computer. All participants provided written informed consent in accordance with the Declaration of Helsinki. In accordance with German law, IRB review was not required for this particular study.

Materials

The study had a 2 × 2 design with set size (small vs large) and distance (short versus long) as factors. To develop the experimental stimuli, verbs were first selected using a corpus and dictionary search of verbs and all their possible particles. Verbs and their particle sets were grouped into small (fewer than six particles) and large (greater than 10 particles) categories and sentences constructed by German native speakers around small/large set pairings. Each experimental item was a quartet of four sentences in which the context required a particle for the sentence to be grammatical. In the example experimental item below, the bolded verb merken (in this context, to note) in (2a/2b) can take only three different particles. Combined with the particle vor (before), its meaning is to take note of or to earmark. In contrast, stellen (to put) in (2c/2d) can take around 18 different particles; when combined with vor (before), its meaning is to introduce. To increase distance between the verb and the particle, we added a long-distance condition where an adjectival modifier was introduced between the verb and its particle (underlined). Crucially, the adjectival modifier did not introduce any new discourse referents or other features that could interfere with the particle’s retrieval (Gibson, 1998; Gibson, 2000; Lewis & Vasishth, 2005). This meant that any slowing due to the additional distance could only be attributed to decay. To balance sentence length between conditions, in the short-distance condition, the intervener was shifted to appear before the verb.

(2) Example item:

a. Small set/short distance:

Nach dem sehr überzeugenden Gespräch merkte er die Kandidatin aus England vor, weil sie ihm sehr gefallen hatte.

Following the very compelling interview, he took note of the candidate from England [particle] because she had really impressed him.

b. Small set/long distance:

Nach dem Gespräch merkte er die sehr überzeugenden Kandidatin aus England vor, weil sie ihm sehr gefallen hatte.

Following the interview, he took note of the very compelling candidate from England [particle] because she had really impressed him.

c. Large set/short distance:

Nach dem sehr überzeugenden Gespräch stellte er die Kandidatin aus England vor, weil sie ihm sehr gefallen hatte.

Following the interview, he introduced the very compelling candidate from England [particle] because she had really impressed him.

d. Large set/long distance:

Nach dem Gespräch stellte er die sehr überzeugenden Kandidatin aus England vor, weil sie ihm sehr gefallen hatte.

Following the interview, he introduced the very compelling candidate from England [particle] because she had really impressed him.

In each experimental item, contexts were matched word-for-word, with the exception of the verb. This was to ensure that the properties of the verb were the only factors contributing to reading times. Ideally, these properties included the number of particles each verb could take. Naturally, it cannot be ruled out that some factor resulting from the internal properties of each verb or its combination with the context contributed to differences in reading times (e.g. taking note of may not generate as narrow an expectation for specific object features as introducing). Furthermore, due to the difficulty of creating sentences with different verbs in matched contexts, it was also not possible to match the frequency of the base verb between conditions. Both of these factors are taken into consideration in interpretation of the results; however, the fact that the base verb is the only word that differs between each sentence gives us the best possible chance to infer that any difference in reading times observed at the particle stem from the verb region of the sentence.

The materials used for the self-paced reading study were 24 items selected from a cloze test, separated into four lists and presented in random order. The lists were compiled using a Latin square design, such that each participant only saw one condition from each item. Each participant therefore saw 24 target sentences, six from each condition, interspersed with 72 filler items. The filler items were either sentences that used particle verbs in other tenses and other syntactic arrangements, or short declarative statements.

Cloze test

In order to confirm that our sentence stimuli (i) elicited particles, (ii) that more particles were elicited by the large set condition than the small set condition, and to (iii) quantify the predictability of the target particle, a cloze test was conducted. An initial total of 48 items, each with four conditions (a–d), was truncated just before the particle such that the verb and the direct object of the sentence were known. German native speakers completed the truncated sentences in a paper-and-pencil cloze test (N = 126, 25 male, mean age 25 years, standard deviation 7 years, range 17–53 years). The 48 sentences were split into four lists such that each participant saw only one condition from every item. The target sentences were randomly interspersed with 63 filler sentences, giving a total of 111 sentences per cloze test. Participants were instructed to complete each truncated sentence with the word or words that first came to mind.

The results of the cloze test yielded 24 items that achieved the required experimental manipulation: a particle was always elicited and more particles were elicited in the large than in the small set condition. It should be noted that in 8% of the stimuli, the highest cloze particle was not used as the target particle. This was because the target particle had to be matched across conditions and the highest cloze particle in one condition was therefore not always the highest cloze particle in another condition. Wherever possible, however, the highest cloze particle was used. Means and 95% confidence intervals of Beta distributions corresponding to the cloze probabilities for each factor level are presented in Table 1.

Table 1 Summary cloze statistics for the final set of 24 items.

Condition	Cloze probability	Entropy	
Mean	95% CI	Mean	95% CI	
Small set	0.51	[0.28, 0.73]	1.10	[1.09, 1.12]	
Large set	0.55	[0.35, 0.75]	1.20	[1.19, 1.22]	
Short distance	0.52	[0.31, 0.73]	1.15	[1.14, 1.16]	
Long distance	0.53	[0.32, 0.75]	1.15	[1.13, 1.16]	
Note:

The 95% CIs reflect confidence intervals of each cloze probability distribution.

Cloze probabilities provided a measure of how predictable the target particles in each condition were. To determine whether the cloze probability of the particle differed between small and large set conditions, a logistic mixed model was fit in brms (Buerkner, 2017) in R (R Development Core Team, 2018) to the cloze probabilities of the target particles, with factor levels contrast coded as follows: small set −0.5/large set 0.5, short distance −0.5/long distance 0.5. The brms zero/one inflated Beta family was used for the likelihood to account for the presence of 0s and 1s in the data. Regularising priors were selected for each of the predictors set size, distance, and their interaction: β ∼ Normal(0, 0.25). The full prior and model specification can be found in the code provided, see “Appendix 1”. The model did not suggest that either set size, distance, or an interaction of the two influenced cloze probability. As can be seen in Fig. 2, the posteriors for the probability of giving the target particle were more or less centred on zero, meaning that neither set size, distance, or their interaction made people any more or less likely to give the target particle.

Figure 2 Change in cloze log odds and entropy of the target particle associated with each predictor.

(A) The posterior distributions for the effect of large set size and long distance on cloze probability relative to the grand mean of each condition (dotted line). The posteriors for the small set size and short distance conditions can therefore be assumed to be the mirror image on the opposite side of the dotted line. The shaded areas are the 95% Bayesian credible intervals. (B) Posteriors for the effect of large set size and long distance on entropy.

The set size manipulation was intended to induce uncertainty about the upcoming particle’s lexical identity; the higher the uncertainty, the less predictable the particle. One useful way of quantifying uncertainty is with entropy. Entropy is a measure of how much information is carried by a new input in light of all possible outcomes.2 In our case, the new input is the particle. In a sentence context where many particles are plausible and cloze probability is uniformly low across all the plausible particles, we assume that uncertainty about the identity of the upcoming particle is high. Thus, each of the plausible particles carries a large amount of information about the meaning of the sentence and entropy is high. In a sentence where only few particles are plausible and one particle is much more probable than the others, we assume that uncertainty about that particle’s identity and the meaning of the sentence is low, and so encountering the high-probability particle will be less informative: this is a low entropy situation.

To determine whether uncertainty (and thus entropy) was higher in the large set condition, a lognormal regression model was fitted to the entropy values with the same contrast coding as for the cloze probability analysis. The brms hurdle lognormal family was used for the likelihood function to account for zeros in the data. Regularising priors were used for the predictors set size, distance, and their interaction: β ∼ Normal(0, 0.01). This model did not suggest that entropy varied with set size, distance, or their interaction, as can be seen in Fig. 2, although the mean entropy was a little higher in the large than the small set condition.

This analysis raised an immediate problem with the experimental design. The categorical predictor set size used in the planned analysis was intended as a proxy for entropy and predictability, where a large set size was supposed to reflect high entropy and thus lower predictability. Although these categories may have reflected the number of particles licensed by each base verb, the results of the cloze test suggested they did not represent the range of particle completions provided by readers at the particle site. This can be seen in Fig. 3: although the average entropy was higher in the large set than in the small set condition, both conditions contained high and low entropy sentences. In other words, there was no difference in predictability of the particle between the small and large set conditions. We therefore present an analysis of entropy as a continuous predictor instead, since this maps better to our planned manipulation of predictability (high entropy = low predictability and vice versa). For transparency, we present both the planned “categorical” analysis and the exploratory “continuous” analysis.

Figure 3 By-item entropy within small and large set categories.

Violin plots show the median and 95% quantiles of the distribution of by-item entropy of the target particle.

Procedure

Participants sat in a quiet booth in the laboratory and read the sentences in 20 point Helvetica font from a 22-inch monitor with 1680 × 1050 screen resolution. Participants saw seven practice items before the experiment proper. The sentences were presented word-by-word in random order using the masked self-paced reading design of Linger (Rohde, 2003). The masked words were presented as underscores separated by spaces. This meant that the participant had some clue as to the length of each word and of the sentence. Participants pressed on the space bar to reveal the next word. The previous word disappeared when the next word appeared, meaning that only one word was visible at any time. Linger recorded the time between word onset and spacebar press, and this data was exported for analysis. After each sentence, a yes/no question appeared which participants answered with the u (No) and r (Yes) keyboard keys. Feedback was not given. The questions concerned the content of the sentences; for example “Was the candidate from America?”. We ensured that the questions targeted a balanced range of sentence regions. A break was offered after every 50 sentences. All other Linger settings were left at their defaults.

Data analysis

Linear mixed models with full variance–covariance matrices estimated for the random effects of participant and item were fitted to the exported Linger data using brms (Buerkner, 2017) in R (R Development Core Team, 2018). Reading times of less than 100 ms were excluded. The dependent variable was reading time at the particle with a 1,000/y reciprocal transform as suggested by the Box Cox procedure (Box & Cox, 1964). We also considered analysing the spillover region, but decided against it as the particle had to be followed by a comma and it was not clear how the clause boundary and associated sentence wrap-up effects (Rayner, Kambe & Duffy, 2000) might interact with reading times in the spillover region. Instead, we present mean reading times across the sentence in Fig. 4. The predictors set size and distance were effect contrast coded: −0.5 (small set/short distance), 0.5 (large set/long distance). The model priors were as follows: β0 ∼ Normal(3, 0.5)

β1,2,3 ∼ Normal(0, 0.5)

υ ∼ Normal(0, συ)

γ ∼ Normal(0, σγ)

συ, σγ ∼ Normal + (0, 0.25)

ρυ, ργ ∼ LKJ(2)

σ ∼ Normal + (0, 0.25)

The prior distribution of the intercept was determined using domain knowledge that mean reading time is approximately three words per second and that 95% of reading speeds should fall within a range of two and four words per second. The slope adjustments, for example β1 (set size), were centred on zero. We assumed that the expected effect of set size would most likely be to either increase or decrease reading speed by, at most, one word per second. By-subject and by-trial adjustments to the slope and intercept (υ, γ) were also centred on zero with respective priors reflecting their plausible standard deviations. The prior for the correlation parameters ρ of these random effects is a so-called LKJ prior in Stan, which takes a hyperparameter η; with an η of two or more, the LKJ prior represents a distribution ranging from −1 to +1, but favours correlations closer to zero. Finally, the prior for the standard deviation parameter σ for the residual is a Normal(0, 0.25) truncated at zero. The full model specification can be found in the code accompanying the article, see “Appendix 1”.

To decide whether the effects of distance and set size were consistent with the null hypothesis that there was no effect, Bayes factors were computed. The Bayes factor gives the ratio of marginal likelihoods for one model against another (Jeffreys, 1939). We therefore compared the planned analysis model including all predictors (described above) against reduced models without the predictor of interest. For example, when we wanted to decide whether the effect of set size was not zero, we computed a Bayes factor for the model with set size (referred to as model 1) vs a reduced model without set size (referred to as model 0): BF10. A Bayes factor of one indicates no evidence in favour of either model. A Bayes factor of greater than 3.0 (when the comparison is BF10) will be taken as evidence in favour of the model with the effect, and a Bayes factor of less than 0.3 as evidence in favour of the null hypothesis. We assessed the strength of the evidence with reference to the conventional Bayes factor classification scheme (Jeffreys, 1939). We computed Bayes factors not only for the planned models, but also for models with more and less informative priors. Computing Bayes factors with a variety of priors is recommended, since the Bayes factor is sensitive to the prior used (Lee & Wagenmakers, 2013).

Results

Question response accuracy and reaction times

Mean accuracy and reaction times to responses to comprehension questions in all four conditions are set out in Table 2.

Table 2 Experiment 1: Summary of question response accuracy and reaction times for comprehension questions.

Condition	Accuracy (%)	Reaction time (ms)	
Mean	95% CI	Mean	95% CI	
(a) Small set, short distance	92	[89, 95]	1,944	[1,862, 2,031]	
(b) Small set, long distance	93	[90, 95]	2,020	[1,918, 2,128]	
(c) Large set, short distance	94	[91, 96]	1,996	[1,897, 2,100]	
(d) Large set, long distance	93	[91, 96]	1,963	[1,872, 2,058]	
Note:

The mean and 95% confidence interval (CI) per condition are presented.

Planned analysis

Set size as a categorical predictor

Mean self-paced reading speed by condition is shown in Table 3 and the model estimates in Table 4. The 95% credible intervals of each of the posteriors contain zero, suggesting that there was uncertainty about how these factors influenced reading speed, if at all. The Bayes factors for all effects were between weakly and strongly in favour of the null hypothesis.

Table 3 Experiment 1: Summary statistics of self-paced reading times by condition using set size as a categorical variable.

Condition	Reading time (ms)	
Mean	95% CI	
(a) Small set, short distance	442	[421, 464]	
(b) Small set, long distance	451	[429, 474]	
(c) Large set, short distance	428	[408, 448]	
(d) Large set, long distance	429	[409, 449]	
Note:

The mean and 95% confidence interval (CI) per condition are presented.

Table 4 Experiment 1: Self-paced reading speed model estimates for the planned analysis with set size as a categorical predictor.

Predictor	β^ (words/s)	95% CrI	Bayes factors (BF10):	
Informative	Planned	Diffuse	
Intercept	2.50	[2.33, 2.67]	–	–	–	
Set size	0.07	[−0.02, 0.16]	1.32	0.28	0.20	
Distance	−0.02	[−0.09, 0.06]	0.31	0.07	0.05	
Set size × distance	0.02	[−0.15, 0.18]	0.88	0.23	0.07	
Note:

The reciprocal transform means that β^ represents the model’s estimated effect for each of the predictors in words per second. A positive sign therefore indicates faster reading (more words per second) and a negative sign, slower reading. The 95% Bayesian credible interval (CrI) gives the range in which 95% of the model’s samples fell. Bayes factors are presented for a range of β priors including, from left to right: more informative than the prior used in the planned analysis, N(0, 0.1); the prior used in the planned analysis, N(0, 0.5); and more diffuse than the prior used in the planned analysis, N(0, 1). BF10 indicates the Bayes factor for the full model (1) against a reduced model (0). Bayes factors of less than 0.3 indicate evidence for the reduced model, while Bayes factors greater than 3.0 indicate evidence for the full model.

Exploratory analysis

Entropy as a continuous predictor

In an exploratory analysis, entropy at the particle was refitted as a continuous predictor and its effect on reading speed examined. Descriptive statistics for reading times in each distance condition are shown in Table 5. Mean reading times according to entropy have been split into high and low categories by median-split for summary purposes, but entropy was used as a continuous predictor in the statistical model.

Table 5 Experiment 1: Summary self-paced reading times by condition using entropy as a continuous variable.

Condition	Reading time (ms)	
Mean	95% CI	
(a) Low entropy, short distance	443	[420, 466]	
(b) Low entropy, long distance	438	[416, 461]	
(c) High entropy, short distance	433	[413, 455]	
(d) High entropy, long distance	443	[422, 466]	
Note:

For the purpose of these summary statistics only, entropy was sorted into high and low categories via median-split. The mean and 95% confidence interval (CI) per condition are presented.

Mean reading times across the whole sentence for both experiments are plotted in Fig. 4. One feature of these data that should be mentioned is that base verbs for sentences with higher entropy at the particle site had a higher corpus frequency than base verbs in sentences with lower entropy at the particle site (to compare verb frequency, we divided sentences into high and low entropy categories via a median split; see Table A1 in “Appendix 2”). Higher corpus frequency of the base verb should have resulted in faster reading times at the verb in high entropy sentences (Kliegl et al., 2004; Rayner & Duffy, 1986), but this was not the case in either experiment. The lack of a frequency effect at the base verb is discussed in the “General Discussion”.

Figure 4 Mean reading times across the sentence in Experiments 1 and 2.

(A and B) Mean self-paced reading times observed in Experiment 1. Error bars show 95% confidence intervals. (C and D) Mean total fixation times observed in Experiment 2.

The priors and model specification remained the same as for the planned analysis. The model coefficients are summarised in Table 6. As can also be seen in Fig. 5, zero is well within the 95% credible interval for the posterior of the all predictors. The Bayes factor analysis found evidence for the null hypothesis for each of the predictors. In other words, there was evidence against an effect of entropy, distance, and their interaction on reading speed.

Table 6 Experiment 1: Self-paced reading speed estimates for the exploratory analysis with entropy as a continuous predictor.

Predictor	β^ (words/s)	95% CrI	Bayes factors (BF10)	
Informative	Planned	Diffuse	
Intercept	2.51	[2.32, 2.69]	–	–	–	
Entropy	−0.04	[−0.13, 0.05]	0.51	0.14	0.07	
Distance	−0.02	[−0.11, 0.07]	0.42	0.10	0.05	
Entropy × distance	−0.02	[−0.15, 0.10]	0.52	0.05	0.01	
Note:

As for the planned analysis, the reciprocal transform means that β^ represents the model’s estimated effect for each of the predictors in words per second. A positive sign therefore indicates faster reading (more words per second) and a negative sign, slower reading. The 95% Bayesian credible interval (CrI) gives the range in which 95% of the model’s samples fell. Bayes factors are presented for a range of β priors including, from left to right: more informative than the prior used in the planned analysis, N(0, 0.1); the prior used in the planned analysis, N(0, 0.5); and more diffuse than the prior used in the planned analysis, N(0, 1). BF10 indicates the Bayes factor for the full model (1) against a reduced model (0). Bayes factors of less than 0.3 indicate evidence for the reduced model, while Bayes factors greater than 3.0 suggest evidence for the full model.

Figure 5 Experiment 1: Change in self-paced reading speed at the particle estimated by the exploratory analysis with entropy as a continuous predictor.

The posterior represents the estimated change in reading time elicited by a one-unit increase in entropy. Due to the reciprocal transform, a shift in the posterior to the left of zero indicates slower reading speeds. The dotted line represents the grand mean of the two factor levels of each predictor and the shaded areas, the 95% credible intervals.

Reading speed predicted by the model is plotted in Fig. 6. The numerical pattern suggests an interesting mix of the two hypotheses: when predictability was high (low entropy), reading speed was faster at long distance in line with the surprisal account. In contrast, when predictability was low (high entropy), the pattern more closely resembles that predicted by decay. However, these patterns are not further interpreted as the outcome of the statistical analysis did not support an interaction.

Figure 6 Experiment 1: Predicted versus modelled self-paced reading times.

(A and B) Predicted interaction patterns in line with surprisal theory and activation decay. (C) Self-paced reading time patterns estimated by the model. Shaded areas indicate 95% Bayesian credible intervals.

Interim discussion

Neither the planned nor the exploratory analyses were consistent with the predictions in Fig. 6. With respect to the planned (categorical) analysis, one potential explanation may lie in the very small differences in cloze probability and entropy at the particle site, meaning that entropy between set size conditions was effectively matched at that point in the sentence. Examples of entropy differences between condition means discussed elsewhere in the literature include 0.38 or 0.50 bits (Levy, 2008), 0.57 bits (Linzen & Jaeger, 2016), and reductions of up to 53 bits (Hale, 2006). In comparison, our between-category difference was only 0.10 bits. However, the examples given from the literature are derived from syntactic entropy of the rest of the sentence, while ours were based on lexical entropy at the particle. Nonetheless, while the small between-category difference in entropy may explain why we did not see a statistical difference in reading times between the large and small set categories, it does not explain why we still saw no difference when entropy was used as a continuous predictor. We turn now to the eye tracking results for further information.

Experiment 2: Eye Tracking

The eye-tracking experiment was conducted using the same materials as the self-paced reading study. Predictability has been shown to affect reading times in both early and total eye tracking measures (Staub, 2015; Rayner, 1998), and the revision of disconfirmed expectations has been associated with a higher rate of regressions (Clifton, Staub & Rayner, 2007; Frazier & Rayner, 1987). Revision of disconfirmed expectations should occur more frequently when predictability is low and the probability of pre-integrating the “wrong” particle increases. We therefore analysed early and total reading times, as well as a measure of regression time. For each of these measures, we maintained the original hypotheses visualised in Fig. 1.

Methods

Participants

Sixty German native speakers were recruited, of which one was excluded due to the presence of a neurological disorder. The remaining 59 (13 male) were free of current or developmental reading or language production disorders, hearing disorders, or vision impairments that could not be corrected without impeding the eye-tracker (e.g. glasses and contacts occasionally caused reflection preventing accurate calibration of the eye-tracker, meaning that these participants had to be excluded if they were unable to read without visual correction). The mean age of the participants was 26 (SD = 6, range = 18–47) and all were university educated. All participants provided written informed consent in accordance with the Declaration of Helsinki. In accordance with German law, IRB review was not required.

Materials

The experimental materials and presentation lists were identical to those used in the self-paced reading study.

Procedure

Right eye monocular tracking was conducted using an EyeLink 1000 eye-tracker (SR Research) with a desktop-mounted camera and a sampling rate of 1,000 Hz. The head was stabilised using a chin and forehead rest which set the eyes at a distance of approximately 66 cm from the presentation monitor. The experimental paradigm was built and presented using Experiment Builder (SR Research). The 22-inch presentation monitor had a screen resolution of 1680 × 1050. Sentences were presented in size 16-point Courier New font on a pale grey background (hex code #cccccc). Each experimental session began with calibration of the eye-tracker, which was repeated if necessary during the experiment. The experimental sentences were preceded by six practice sentences. Participants fixated on a dot at the centre left of the screen before each sentence was presented. Once they had finished reading, they fixated on a dot at the bottom right of the screen. Each of the experimental sentences was followed by the same yes/no question used in the self-paced reading study, which the participant answered using a gamepad. Each session lasted approximately 30 min.

Data analysis

Sampled data were exported from DataViewer (SR Research) and pre-processed in R using the em2 package (Logačev & Vasishth, 2013). Trials containing blinks or track loss were excluded. Linear mixed-effects models with full variance–covariance matrices estimated for the random effects of participant and item were fitted using brms (Buerkner, 2017) in R (R Development Core Team, 2018) separately to data for each of four reading time measures: first fixation duration, first pass reading time, total fixation time, and regression path duration. This range of measures was selected as both early and late measures have been found to be affected by predictability (Kliegl et al., 2004; Boston et al., 2008), although perhaps earlier measures are more sensitive (Staub, 2015). The target region of the sentence was the particle plus the immediately preceding word, since the particles were usually short (two to three letters) and therefore not always fixated. As for Experiment 1, the spillover region was not analysed, but mean reading times across the whole sentence are presented in Fig. 4. The preceding rather than the following word was chosen because the target particle was at the right clause boundary. The dependent variables were first fixation duration, first pass reading time, total fixation time, and regression path duration at the particle, log transformed as indicated by the Box Cox procedure. The predictors set size and distance were effect contrast coded: −0.5 (small set/short distance), 0.5 (large set/long distance). The model priors were as follows: β0 ∼ Normal(5.7, 0.5)

β1,2,3 ∼ Normal(0, 0.5)

υ ∼ Normal(0, συ)

γ ∼ Normal(0, σγ)

συ, σγ ∼ Normal+ (0, 1)

ρυ, ργ ∼ LKJ(2)

σ ∼ Normal+ (0, 1)

The prior distribution of the intercept was determined using domain knowledge that mean reading time is approximately 300 ms (5.7 on the log scale) and that 95% of reading times should fall within a range of 110 and 812 ms. We expected the effect of the predictors would mostly lie somewhere between a speed-up of 190 ms and a slow-down of 513 ms. Priors for the random effects parameters were as shown above. The full model specification can be found in the code in the accompanying code, see “Appendix 1”.

Results

Question response accuracy and reaction times

Mean response accuracy and reaction times for the comprehension questions in all four conditions are set out in Table 7.

Table 7 Experiment 2: Summary of question response accuracy and reaction times.

Condition	Accuracy (%)	Reaction time (ms)	
Mean	95% CI	Mean	95% CI	
(a) Small set, short distance	91	[88, 94]	2,052	[1,967, 2,141]	
(b) Small set, long distance	92	[89, 95]	2,090	[2,007, 2,177]	
(c) Large set, short distance	96	[94, 98]	2,007	[1,928, 2,089]	
(d) Large set, long distance	97	[94, 98]	2,051	[1,978, 2,126]	
Note:

The mean and 95% confidence interval (CI) per condition are presented.

Planned analysis

Set size as a categorical predictor

Observed reading times per condition are summarised in Table 8. The model estimates for each reading time measure are shown in Table 9. The 95% credible interval for each of the posteriors contains zero, suggesting that it was uncertain whether the predictors’ effect on any reading time was positive or negative, or zero. However, as for the self-paced reading experiment (Experiment 1), the categorical distinction of large and small set size was probably inappropriate, and thus an exploratory analysis using entropy as a continuous predictor is presented next. A possible limitation of our approach using Bayes factor analyses is that we are evaluating multiple measures, without any correction for family-wise error (von der Malsburg & Angele, 2016). While the family-wise error rate is a frequentist concept, it may be that an analogous issue exists in the Bayesian framework for which we have not controlled. Our analyses should therefore be considered exploratory and confirmed via future replication attempts.

Table 8 Experiment 2: Summary statistics of eye-tracking reading times by condition using set size as a categorical variable.

Measure	Condition	Reading time (ms)	
Mean	95% CI	
First fixation duration	(a) Small set, short distance	284	[269, 299]	
	(b) Small set, long distance	285	[270, 301]	
	(c) Large set, short distance	292	[277, 309]	
	(d) Large set, long distance	303	[287, 319]	
First pass reading time	(a) Small set, short distance	316	[297, 335]	
	(b) Small set, long distance	313	[294, 333]	
	(c) Large set, short distance	324	[304, 345]	
	(d) Large set, long distance	337	[317, 357]	
Total fixation time	(a) Small set, short distance	368	[343, 395]	
	(b) Small set, long distance	364	[338, 391]	
	(c) Large set, short distance	370	[344, 397]	
	(d) Large set, long distance	381	[355, 408]	
Regression path duration	(a) Small set, short distance	354	[330, 379]	
	(b) Small set, long distance	355	[330, 382]	
	(c) Large set, short distance	359	[334, 386]	
	(d) Large set, long distance	380	[354, 408]	
Note:

The mean and 95% confidence interval (CI) per condition are presented.

Table 9 Experiment 2: Model estimates for the planned analysis with set size as a categorical predictor.

Measure	Predictor	β^ (log ms)	95% CrI	Bayes factors (BF10)	
Informative	Planned	Diffuse	
First fixation duration	
	Intercept	5.66	[5.55, 5.75]	–	–	–	
	Set size	0.02	[−0.01, 0.05]	1.69	0.10	0.02	
	Distance	0.01	[−0.02, 0.03]	0.27	0.06	0.04	
	Set size × distance	0.01	[−0.02, 0.03]	0.19	0.00	0.00	
First pass reading time	
	Intercept	5.74	[5.58, 5.89]	–	–	–	
	Set size	0.02	[−0.01, 0.05]	2.02	0.10	0.02	
	Distance	0.00	[−0.02, 0.03]	0.27	0.05	0.03	
	Set size × distance	0.01	[−0.02, 0.03]	0.32	0.01	0.00	
Total fixation time	
	Intercept	5.89	[5.71, 6.06]	–	–	–	
	Set size	0.00	[−0.04, 0.04]	1.16	0.09	0.02	
	Distance	0.00	[−0.03, 0.03]	0.28	0.05	0.03	
	Set size × distance	0.01	[−0.04, 0.04]	0.59	0.02	0.00	
Regression path duration	
	Intercept	5.86	[5.69, 6.03]	–	–	–	
	Set size	0.01	[−0.03, 0.05]	1.38	0.08	0.02	
	Distance	0.01	[−0.02, 0.04]	0.41	0.07	0.04	
	Set size × distance	0.01	[−0.02, 0.04]	0.80	0.05	0.01	
Note:

β^ represents the model’s estimated effect for each of the predictors on the log scale. The log transform means that estimates with a positive sign indicate slower reading times and that readers who are slower on average will be more affected by the manipulation than faster readers. The 95% Bayesian credible interval (CrI) gives the range in which 95% of the model’s samples fell. Bayes factors are presented for a range of β priors including, from left to right: more informative than the prior used in the planned analysis, N(0, 0.1); the prior used in the planned analysis, N(0, 0.5); and more diffuse than the prior used in the planned analysis, N(0, 1). BF10 indicates the Bayes factor for the full model (1) against a reduced model (0). Bayes factors of less than 0.3 indicate evidence for the reduced model, while Bayes factors greater than 3.0 suggest evidence for the full model.

Exploratory analyses

Entropy as a continuous predictor

As for the self-paced reading analysis, models were refit using entropy as a continuous predictor. Descriptive statistics for each reading time measure are shown in Table 10. Mean reading times according to entropy have been split into high and low categories by median-split for summary purposes, but entropy was used as a continuous predictor in the statistical model.

Table 10 Experiment 2: Summary eye-tracking reading times by condition using entropy as a continuous variable.

Measure	Condition	Reading time (ms)	
	Mean	95% CI	
First fixation duration	
	(a) Low entropy, short distance	279	[265, 295]	
	(b) Low entropy, long distance	264	[250, 279]	
	(c) High entropy, short distance	293	[277, 311]	
	(d) High entropy, long distance	317	[299, 335]	
First pass reading time	
	(a) Low entropy, short distance	317	[297, 338]	
	(b) Low entropy, long distance	287	[270, 306]	
	(c) High entropy, short distance	321	[300, 343]	
	(d) High entropy, long distance	357	[334, 381]	
Total fixation time	
	(a) Low entropy, short distance	357	[332, 385]	
	(b) Low entropy, long distance	321	[299, 346]	
	(c) High entropy, short distance	376	[348, 407]	
	(d) High entropy, long distance	416	[385, 449]	
Regression path duration	
	(a) Low entropy, short distance	354	[329, 382]	
	(b) Low entropy, long distance	325	[301, 351]	
	(c) High entropy, short distance	358	[332, 386]	
	(d) High entropy, long distance	402	[373, 433]	
Note:

For the purpose of these summary statistics only, entropy was sorted into high and low categories via median-split. The mean and 95% confidence interval (CI) per condition are presented

The model estimates can be seen in Table 11 and the model posteriors in Fig. 7. The Bayes factor analysis found evidence for an effect of entropy on first fixation duration, first pass reading time, and total fixation time, in that increasing entropy slowed reading times. With more informative priors, Bayes factors suggested evidence for the effect of entropy in each of these three measures was strong. At the planned (non-informative, regularising) prior for regression path duration, Bayes factor evidence for an effect of entropy was inconclusive. However, when the more informative prior was used, evidence for an effect of entropy on regression path duration was strong. The Bayes factors for the remaining predictors (distance, entropy × distance) were in favour of the null hypothesis, regardless of which prior was used.

Table 11 Experiment 2: Model estimates for the exploratory analysis with entropy as a continuous predictor.

Measure	Predictor	β^ (log ms)	95% CrI	Bayes factors (BF10)	
Informative	Planned	Diffuse	
First fixation duration	
	Intercept	5.66	[5.55, 5.76]	–	–	–	
	Entropy	0.08	[0.03, 0.13]	23.88	4.65	2.15	
	Distance	0.01	[−0.05, 0.07]	0.28	0.06	0.03	
	Entropy × distance	0.04	[−0.04, 0.11]	0.32	0.01	0.00	
First pass reading time	
	Intercept	5.76	[5.61, 5.90]	–	–	–	
	Entropy	0.08	[0.03, 0.13]	17.71	4.49	1.86	
	Distance	0.00	[−0.06, 0.07]	0.27	0.06	0.03	
	Entropy × distance	0.02	[−0.06, 0.10	0.19	0.00	0.00	
Total fixation time	
	Intercept	5.87	[5.70, 6.04]	–	–	–	
	Entropy	0.12	[0.04, 0.21]	24.65	4.77	2.78	
	Distance	0.00	[−0.06, 0.07]	0.32	0.07	0.04	
	Entropy × distance	0.01	[−0.08, 0.09]	0.22	0.00	0.00	
Regression path duration	
	Intercept	5.85	[5.67, 6.02]	–	–	–	
	Entropy	0.10	[0.03, 0.18]	12.58	2.91	1.18	
	Distance	0.01	[−0.05, 0.08]	0.35	0.07	0.03	
	Entropy × distance	0.04	[−0.06, 0.12]	0.41	0.01	0.00	
Note:

β^ represents the model’s estimated effect for each of the predictors on the log scale. The log transform means that estimates with a positive sign indicate slower reading times and that readers who are slower on average will be more affected by the manipulation than faster readers. The 95% Bayesian credible interval (CrI) gives the range in which 95% of the model’s samples fell. Bayes factors are presented for a range of β priors including, from left to right: more informative than the prior used in the planned analysis, N(0, 0.1); the prior used in the planned analysis, N(0, 0.5); and more diffuse than the prior used in the planned analysis, N(0, 1). BF10 indicates the Bayes factor for the full model (1) against a reduced model (0). Bayes factors of less than 0.3 indicate evidence for the reduced model, while Bayes factors greater than 3.0 suggest evidence for the full model.

Figure 7 Experiment 2: Changes in reading time for each eye-tracking measure using entropy as a continuous predictor.

(A–D) The posterior represents the estimated change in reading time for the average reader elicited by a one-unit increase in entropy. The log transformed reading times mean that posteriors shifted to the right of zero indicate slower reading. Shaded areas indicate the 95% Bayesian credible intervals.

The predicted vs observed interactions of distance and entropy are plotted in Fig. 8. Numerically, the pattern of reading times again appeared to be a mixture of the predictions of surprisal theory and the decay simulation based on the LV05 model. However, the results of the statistical analyses did not support an interaction of entropy and distance, and so this pattern is not further interpreted.

Figure 8 Experiment 2: Predicted vs modelled interaction of entropy and distance on reading times in each eye tracking measure.

FFD refers to first fixation duration, FPRT to first pass reading time, TFT to total fixation time, and RPD to regression path duration. (A and B) Predicted interaction patterns in line with surprisal theory and activation decay. (C–F) Observed reading time patterns. Shaded areas represent 95% Bayesian credible intervals.

Interim discussion

The planned analysis with the categorical predictor set size again did not find any support for our hypotheses that temporal activation decay would be more prominent when lexical predictability was low. Reconfiguring set size as the continuous predictor entropy, however, found support for the hypothesis that increased uncertainty about the lexical identity of the particle would slow reading times. However, there was still no evidence that temporal decay influenced reading times, either alone or in interaction with entropy.

General Discussion

In two reading time experiments, we investigated whether readers pre-activated the lexical identity of a particle in long-distance verb-particle dependencies by varying lexical predictability of the particle. We additionally examined whether delaying the appearance of the particle would facilitate processing in line with the surprisal account (Levy, 2008), whether processing might be negatively affected by temporal activation decay, and whether the particle’s lexical predictability might interact with either of these factors. The planned analyses of both a self-paced reading and an eye tracking experiment provided evidence against an effect of particle predictability or delay of its appearance. However, in more appropriate exploratory analyses using entropy as a continuous predictor at the particle site, we did find evidence of particle predictability in eye-tracking but not self-paced reading, and evidence against an effect of decay or its interaction with predictability in any modality.

The findings in the eye tracking data are consistent with evidence suggesting that the effects of predictability influence early stages of lexical processing and thus that its effects are more likely to be detected in early eye tracking measures (Staub, 2015), as well as gaze duration (Rayner, 1998). At first blush, our results appear inconsistent with this proposal in that we observed a predictability effect in both early and late eye tracking measures, including regression path duration. However, this may have been due to the fact that first fixation durations were included in the computation of the remaining three measures, meaning that the primary source of the effect may actually be first fixation durations (Vasishth, von der Malsburg & Engelmann, 2013). On the other hand, it is possible that regression path duration times may reflect the reanalysis of a mispredicted particle in the high entropy (low predictability) sentences, rather than faster early lexical access in low entropy (high predictability) sentences (Clifton, Staub & Rayner, 2007; Frazier & Rayner, 1987). Our design does not enable us to distinguish between these two possibilities, but either mechanism is consistent with pre-activation of the long-distance particle.

When was the particle pre-activated?

Within each experimental item, all words were identical except for the verb, meaning that the only information influencing uncertainty at the particle site was the verb. This supports the possibility that the difference in reading time observed at the particle could have resulted from differences in particle pre-activation at the verb. However, it is also possible that pre-activation was triggered by the combination of the verb and its direct objects. For example the fragment ‘Nach dem Gespräch stellte er die Kandidatin…’ (Following the interview, he put the candidate…) should be sufficient to anticipate the most likely verb-particle combinations. The lexical pre-activation of particles is unlikely to have been triggered by information between the direct object and the particle site (e.g. ‘aus England’, from England), since this region did not add any information about the identity of the particle. It is therefore possible to conclude that pre-activation occurred at the latest before the pre-critical region, suggesting that lexical pre-activation can be sustained over multiple intervening words that do not form part of the particle verb constituent (cf. studies where evidence for lexical pre-activation is only observed at the immediately preceding word or within the noun phrase: DeLong, Urbach & Kutas, 2005; Van Berkum et al., 2005; Wicha, Moreno & Kutas, 2004; Nicenboim, Vasishth & Rösler, 2020).

One feature of interest in the data, and perhaps in further support of particle pre-activation at the verb, is the fact that base verbs associated with higher entropy at the particle were higher in frequency, and yet were not read faster. High word frequency is strongly correlated with faster reading time (Kliegl et al., 2004; Rayner & Duffy, 1986). A potential explanation for the lack of a speed-up is that a larger number of pre-activated particles made the meaning of the verb more ambiguous, which in turn led to slower reading and cancelling out of the expected speed-up associated with higher frequency. This hypothesis requires testing, however.

Assuming that particle pre-activation underlies the effects observed in eye-tracking, our findings present a contradiction to the hypothesis that verbs that take particles are maintained in working memory to facilitate retrieval once the particle is finally encountered (Piai et al., 2013). If this were the case, we should not have observed an effect of predictability at the particle, since there is no reason to think that one verb, already activated and integrated into the sentence parse, should have required more resources to retrieve than another. It may indeed be that high entropy verbs are somehow more difficult to integrate than low entropy verbs, but it is difficult to conceive of why without invoking activation of associated lexical or syntactic information, including particles. Maintenance of the verb in working memory therefore does not account for the eye-tracking results observed reported here.

Temporal activation decay

The evidence against an effect of temporal decay in both the self-paced reading and eye tracking experiments is consistent with findings suggesting that decay is not an important factor influencing reading and memory recall times (Lewandowsky, Oberauer & Brown, 2009; Engelmann, Jäger & Vasishth, 2019; Vasishth et al., 2019). In comparison to the sentences used in distance manipulations in previous studies, our sentences used simple adjectival modifiers that deliberately avoided the introduction of interference or new discourse referents. This allowed us to isolate decay as an explanatory factor; however, it is possible that the modifiers were not long enough to introduce a detectable effect of decay. That said, it would have been difficult to construct longer interveners without reintroducing interference or working memory load, which supports the idea that interference and working memory load are indeed the more important source of processing difficulty in longer sentences, rather than temporal decay. Alternatively, it could be argued that the difficulty in constructing longer sentences without introducing interference or working memory load means it is difficult or impossible to test decay in isolation, and thus that we cannot know what the true effect of decay is. However, if the effect of decay is so small that it is undetectable in the face of interference and working memory load, and these factors are almost unavoidable in constructing long dependencies, then one could argue that decay does not play a major role in processing difficulty.

Another possible explanation for not having detected a decay effect is that the difficulty in creating experimental items meant there were only 24 experimental items in total. In the Latin square design, this meant that each participant saw only six target trials per condition. If the effect of decay is indeed very small, future experiments should include more trials per participant in order to detect the effect.

Conclusions

We investigated whether readers pre-activate the lexical content of long-distance verb-particle dependencies such as turn the music down, or whether they wait to interpret the meaning of the verb retrospectively once the particle is encountered. In addition, we compared two hypotheses of dependency processing: whether delaying the appearance of a verb particle would facilitate its processing (an antilocality effect), or whether activation decay over time would negatively impact its processing (a locality effect). We found evidence that readers did pre-activate the lexical identity of upcoming particles and that this pre-activation facilitated early processing stages, but evidence against any effect of delaying the particle on processing. Crucially, the particle in the current study was delayed with information that neither hinted at the upcoming particle’s identity, nor increased interference or working memory load. The evidence against an effect of delaying the particle therefore suggests that locality and antilocality effects observed in previous research may be due to the additional intervening information that adds to working memory load or confirms lexical expectations, and that temporal activation decay is not a strong influence on reading times.

Appendix 1

Data and code

All data and code necessary to reproduce our analyses are available here: https://osf.io/yg5wx/.

Appendix 2

Particle verb frequencies

Frequencies were computed for both the base verb and the particle verb as a whole using the Tübingen aNotated Data Retrieval Application, TüNDRA (Martens, 2013). The treebank used was the automatic dependency parse of the German Wikipedia with over 48.26 million sentences. Frequencies are presented as the incidence of the verb or particle verb per 1,000 words. As can be seen in Table A1, while the frequencies of the verb+particle constructions were comparable, frequency of the base verb was notably higher in the high entropy condition.

10.7717/peerj.10438/table-A1 Table A1 Mean verb and particle verb frequency per 1,000 words for high and low entropy.

Sentences were divided into high and low entropy categories via a median split.

Condition	Verb only	Verb+particle	
Mean	95% CI	Mean	95% CI	
Low entropy	0.17	[0.11, 0.28]	0.04	[0.03, 0.07]	
High entropy	0.42	[0.26, 0.69]	0.04	[0.03, 0.07]	

We thank Genevieve McArthur, Aya Meltzer-Asscher, Stefan Frank, and Michael Hahn for their detailed and thoughtful reviews of the manuscript, which greatly improved the final paper. We also wish to acknowledge the valuable feedback from reviewers and colleagues at AMLaP 2017 and CUNY 2018.

Additional Information and Declarations

Competing Interests

Author Contributions

Human Ethics

Data Availability

1 We attempted to compute surprisal values using the Incremental Top-Down Parser (Roark & Bachrach, 2009) and two different types of annotated corpora: the Tiger newspaper corpus (Brants et al., 2004); and a larger corpus of novels annotated with the German version of the Stanford CoreNLP natural language software (Manning et al., 2014). However, the particles were incorrectly categorised by the parser (e.g. as adverbs, verbs, and even nouns), making the surprisal estimates unreliable.

2 Entropy (H) was calculated as the negative sum of cloze probabilities (P) for all particles provided by participants for a particular sentence in the cloze test, multiplied by their respective logs: H=−∑iPilog2⁡Pi. For example if nine cloze completions were the particle “vor" and one was “an”, then: H = − (Pvor ⋅ log2 Pvor + Pan ⋅ log2 Pan) = −(0.9 ⋅ log2 0.9 + 0.1 ⋅ log2 0.1) = 0.47.

Shravan Vasishth is an Academic Editor for PeerJ.

Kate Stone conceived and designed the experiments, performed the experiments, analyzed the data, prepared figures and/or tables, authored or reviewed drafts of the paper, and approved the final draft.

Titus von der Malsburg analyzed the data, authored or reviewed drafts of the paper, and approved the final draft.

Shravan Vasishth conceived and designed the experiments, authored or reviewed drafts of the paper, and approved the final draft.

The following information was supplied relating to ethical approvals (i.e., approving body and any reference numbers):

The study was conducted in line with the Declaration of Helsinki. In accordance with German law, IRB review was not required.

The following information was supplied regarding data availability:

Data and code are available on the Open Science Framework repository: Stone K, von der Malsburg T, Vasishth S. 2020. The effect of decay and lexical uncertainty on processing long-distance dependencies in reading. Available at osf.io/yg5wx. DOI 10.17605/OSF.IO/YG5WX.

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
