# Peer review of "The effect of decay and lexical uncertainty on processing long-distance dependencies in reading"

_PeerJ, doi:10.7717/peerj.10438_

## Round 0.1 · original submission · Major Revisions

Dear Dr/Profs Stone, von der Malsburg, and Vasishth,

Thank you for submitting your article "The effect of decay and lexical uncertainty on processing long-distance dependencies in reading" to PeerJ. At the outset, I must apologise for my delayed response to you. All three reviewers provided their reviews in good time. However, Australia has its summer holidays in the late December/early January period and hence I was on leave when the reviews arrived. I returned to work yesterday, and have prioritised your paper as quickly as possible.

As mentioned, three reviewers have had a close look at your manuscript, and all provide favourable responses. One reviewer suggests accepting the manuscript as is; two reviewers have asked for further clarification about various aspects of the manuscript to improve understanding. Since it is important for both yourselves, as well as the journal, to make the content of your research and manuscript accessible to as many readers as is possible, I suggest that you address all the comments provided by Reviewers 1 and 2 - either in the manuscript itself (ideally) or in a response to the appropriate reviewer (if the clarification/suggestion cannot be accommodated in the manuscript). Reviewer 3 also makes some useful suggestions for improvement. While the reviewer indicates that these suggestions are not mandatory, for the sakes of accessibility, I suggest your address as many of these in the manuscript as is possible, and again provide a direct response to the reviewer if this is not possible.

I believe that the suggested clarifications are important in light of my own review of the manuscript. Like the reviewers, I believe the article is well written - particular for a first submission. However, the research is specialised and quite "dense", and hence would benefit from making it easier to understand for non-specialists. To this end, I would suggest the following (minor) changes in addition to those outlined by the three reviewers:

(1) Predictions section (page 3). I found this section confusing. It seemed to "come out of the blue" - partly due to unclear wording, I believe, and partly because not much background had been provided about the two models that were being pitched against each other. I believe Reviewer 2 had a similar concern, and has offered some specific suggestions for how this might be addressed in the Introduction. In addition to those suggestions, please ensure that the Prediction section is precluded by a clear explanation of the two theories, and that the logic behind each prediction is described as clearly and simply as possible.

(2) Participants section (page 4). Please clarify if "language" disorders include reading disorders.

(3) Materials section (page 4). I was a bit confused by the presentation of the stimuli. Would it be possible to reformat the examples to improve clarity by adding a blank line between the two lines of the German/English stimuli, and also provide the meaning of the text prior to the stimuli? For example, something like:

Small set/short distance (perhaps in bold)

Meaning: With the newly bought rag, she scrubbed the plates in the kitchen to create space for cooking

German: Mit dem ....
English: With the ....

German: Platz zum ....
English: Place for ....

(I hope that makes sense).

(4) I understand why you might decide to outline the history of the development of the stimuli under Materials (pages 4-7). However, the length of this history narrative the reader from the flow of information for Experiment. I wonder if this extra information might be included in a supplementary file OR described as a separate experiment prior to Experiment 1 and 2.

(5) At some point, there appeared to be an abrupt switch from the use of the term "predictability" to "entropy". I think I worked out that they were related concepts, but I could not tell if they were the same thing, given this area of research is not my area of expertise. If they are the same thing, it would help the reader to use the term "predictability" throughout the manuscript, since it is a less specialised word. However, if a switch to entropy is required, this needs to be explained clearly at the appropriate point in the narrative.

I hope you find the suggestions of the reviewers, plus my own minor comments, useful in the further development of your work.

Genevieve

·

Basic reporting

Excellent reporting, although minor improvements are possible (see General Comments)

Experimental design

The only thing that remains unclear is if/how spillover was taken into account (see General Comments)

Validity of the findings

No comment

Additional comments

This paper deals with a topic that is very timely and relevant to the study of human sentence processing. The experiments are well designed, the analyses are state of the art, and the writing is very clear. Nevertheless, there are two issues need to be resolved before I can recommend publication.
(1) Does the verb in every stimuli sentence require a particle? If not, how was the “no particle” option incorporated in the cloze test and data analyses?
(2) There is no mention of how spillover was taken into account, even though this phenomenon is prevalent in reading, in particular self-paced reading. Were reading times on words directly following the particle also taken considered? If not, could this be why the expected effects were not found?

Minor comments:
- line 47: what does it mean for something to be “anecdotally assumed”?
- When introduction German particle verbs, it would be good to mention that moving the particle to after the object NP is required in German.
- line 115: the Dutch prefix “ver” in “verdelen” is not a particle (i.e., it is not split: “hij deelt het ver” is not possible)
- line 127-128: “self-paced reading and eye tracking modalities” and “reading modalities” -> shouldn’t this be “paradigm” instead of “modality”? In both cases, the modality is written/visual.
- Table 1 shows 95% CI instead of the standard error mentioned on line 221. Also, the caption is not quite accurate because the table presents cloze statistics but not the cloze test results.
- It would be helpful if the goal of the cloze test data analysis were explained before the technical details (starting on line 226)
- line 232-234: “the probability of the target particle was lower … for the interaction” -> for which combination of factor levels was the probability lower?
- The violin plots of Fig. 4 shows probability mass for negative values of entropy, even though entropy is by definition non-negative
- line 414: what did the preprocessing of eye-tracking data entail?
- line 417: the citation to R is “R Core Team”, not just “Team”.
- line 448-449: the problem of evaluating multiple dependent measures is not a “limitation of the BF analysis” in particular, is it?
- line 474: “The statistical analysis” should probably be “The outcome of the statistical analysis”

·

Basic reporting

No comment (see section 4 below for all comments).

Experimental design

No comment

Validity of the findings

No comment

Additional comments

The paper reports the results of one self-paced reading and one eyetracking-while-reading experiments, aimed to investigate the effect of decay, and its interaction with predictability, on the processing of verb particles which are dependent on the verb but appear downstream from it. This is an interesting question, the experiments are overall well thought-out, the analysis is rigorous (with data and code provided) and the discussion is careful and responsible.
In my view, the main weakness of the manuscript is the presentation the hypotheses of the two frameworks, and in particular that of the LV05 model. Personally, the reasons for the predictions of the LV05 model were not clear to me, which made it hard to understand certain aspects of the experiment and interpret the results. Below I elaborate some more on this concern, and offer some other, more minor, comments.

Clarity of opposing hypotheses, particularly the predictions of the LV05 model:

To me, the Introduction (particularly the first page) was very confusing.
If I understand correctly, the experiments set out to test the predictions of Surprisal vs. LV05. First, I think this should be stated clearly and consistently throughout the paper, starting from the abstract (where now only Surprisal is mentioned, in contrast to "other theories") and then explicitly in the Introduction. (And also in the discussions - the Discussion of the SPR results now starts with "we hypothesized" – is "we" = LV05? and the hypothesis of Surprisal is not spelled out at all there; the authors only state that the results were not consistent with it).
Once this is established, I think the authors should take MUCH more time to introduce these frameworks. Currently, both Surprisal and the LV05 model are simply mentioned (along with their relevant predictions for the current research, but with no other explanation). I think the authors should present these theories for readers who are unfamiliar with them. What is Surprisal theory, what are its main tenets? What is the LV05 model? What is it modelling, what are its assumptions?
Then, the authors should explain both frameworks' hypotheses about decay and its interaction with predictability. For Surprisal, line 40 "interference and working memory constraints may negatively impact…": in "working memory constraints", do the authors mean "the limited capacity of working memory"? Why would interference and limited capacity interact with predictability in this way, according to Surprisal? More explanation is needed, along with the relevant results (from German? Hindi? Persian?)
Even more so, for the LV05 model, it was hard to understand the proposed reason for the interaction between predictability and decay. The crucial sentence is this: "If an upcoming lexical item is highly predictable, it can be pre-integrated into the pursued parse, facilitating its retrieval once encountered. However, if there is uncertainty about the lexical identity of a word, this will increase the likelihood that the parser either pursues a parse with a different lexical item to the one yet to be encountered, or makes no lexical prediction at all".
This raised a lot of questions for me:
- Line 149 onwards "in the absence of interference, decay over distance … will make the long condition more sensitive to predictability". Why? I understand that these are the results of a simulation, but can the authors provide an intuition as to why this is so? Do the authors claim that when a lexical item is highly predictable, it is integrated (prior to its occurrence in the input) and it is therefore amenable to decay? If so, it should be stated clearly.
- What's "highly predictable"? Consider for example a verb from the small set size group which takes five possible particles. If one of them appears in 80% of cases, and each of the other four – in 5% of cases, is the most probable one highly predictable, therefore integrated and amenable to decay? If this is the case, shouldn't that also happen for a hypothetical verb in the large set size group which takes 15 particles, with the most probable one appearing in 80% of cases, and each of the other ones in ~1.5% of cases? And what about a "small set" verb with 60%-10%-10%-10%-10% distribution of particles and a "large set" verb with 60%-4%-4%... distribution? Would the most probable particle be integrated?
- What happens when there's no one highly predictable completion? What's the role of decay in these cases? What's the predicted difference between a small set verb with 5 possible completions each appearing in 20% of the time, and a large set verb with 15 completions, one appearing in 20% of the time and each of the others in ~6%?
- The upshot from the last two questions is: shouldn't we look at constraint (cloze probabilities) *at the verb* in order to know what was preactivated/integrated there? Or perhaps at entropy, if it is assumed to modulate preactivation/integration (e.g. integration only happens when there are no strong competitors, i.e. low entropy), but again, *at the verb*? As the authors say in line 263, the study wanted to test "whether the number of potential particles pre-activated at the verb would affect reading times". But to know whether they are preactivated, don't we need cloze data from that point? (even though possibly subsequent material, i.e. the object, can prove our prediction wrong, leading to reanalysis? Since I'm not sure what the assumptions of LV05 are, I don't know what it will predict).
- The manuscript does discuss entropy, but measured right before the particle. In the pre-test, it turns out that there's no difference between the two groups, but this is only discussed in the Results section, before carrying out the alternative analysis. I think it would be much better to acknowledge the potential problem, namely that the two verb types have similar entropies (before the particle), and why this may undermine the verb type manipulation, when the pretest is presented. Otherwise the reader is left very confused.
- This is related to another minor point that was not clear to me: how were the verbs selected? Based on the cloze pretest, namely based on their preference after the object, before the particle? Or based on their particle selection options regardless of the specific object?

Other comments:
Line 36, after "there are accounts modeling the effect of intervening material…": I think it is natural to start the Introduction with the discussion of decay (which now appears in the second paragraph), as these are the more traditional approaches to distance effects. Then, Surprisal and anti-locality can be presented.
The manipulation of decay was introduced by adding a very short constituent – a two-word phrase. Could that be the reason why no effect of decay was found? Does the LV05 predict an effect of decay with such a minimal manipulation? Related to this, line 526, "it would have been difficult to construct longer sentences without reintroducing these factors (interference), which supports the idea that they are the source of processing difficulty": why does it support this idea? I think it only means that it's very hard (perhaps impossible?) to test the influence of decay by itself.
When entropy is first discussed, the concept should be explained – not only with a mathematical formula, but also with the intuition as to what it means.
Minor comments
Line 47 "activation decay is anecdotally assumed…": another relevant reference here is Chow & Zhou (2019), which is a replication of Wagers & Phillips (2014) (though the original authors do not frame their study as investigating decay).
Line 52 "decay is not a useful predictor": perhaps also cite Van Dyke and Johns' (2012) review which argues against a role for decay in sentence processing.
Materials section: Do all the experimental verbs necessarily take particles at all? I assume this is the case, but I think this should be stated explicitly
Line 217 "24 items that suited the experimental design" – meaning what? That they selected 6 or less, or 15 or more, particles?
Online norming study (line 249 onwards): Why is this pretest necessary? In the experiment, the verb is several words upstream from the particle, so why are reading times of the verb+particle relevant?
Line 382, "a second possibility is that locality and antilocality effects simply cancelled each other out": how is this relevant to the effect of predictability, which is the topic of discussion? I would think that it is relevant to the (lack of) effect of decay, not predictability.
Line 484 "speed up at the verb": this sounded to me like the authors were referring to a speed up at the verb relative to preceding material; it took me some time to understand that it means lower reading times in the large set verbs compared to the small set verbs.
Line 544, "a potential explanation for the lack of speed-up… more preactivated particles may have led to slower reading". I'm not sure I would predict this. I would think activations are not usually viewed as costly. Perhaps the source of increased reading times here is that the verbs are more ambiguous/vague, i.e. have more possible meanings?
Typos etc.
line 34: length > amount
line 166: items > item
line 128: delete second 'also'
line 319: delete second 'the'
line 341 and caption for Figure 5: I initially thought the RTs in the table are reading times for the particle (and wondered why they were so high). The text and caption should say that these are RTs for answering the comprehension questions. Same for line 440 and table 9.
Line 389: the number "1" is missing
Line 457, "the results of the statistical analysis": in all the reading time measures? If so, maybe "analyses"?

References:
Chow, W. Y., & Zhou, Y. (2019). Eye-tracking evidence for active gap-filling regardless of dependency length. Quarterly Journal of Experimental Psychology, 72(6), 1297-1307.‏
Van Dyke, J. A., & Johns, C. L. (2012). Memory interference as a determinant of language comprehension. Language and linguistics compass, 6(4), 193-211.‏
Wagers, M. W., & Phillips, C. (2014). Going the distance: Memory and control processes in active dependency construction. The Quarterly Journal of Experimental Psychology, 67(7), 1274-1304.‏

·

Basic reporting

The paper is well-written and clear. References and contextualization are appropriate, and the paper is self-contained. Raw data is available.

Comments:
Supplemental material is referenced (e.g., Page 8, last sentence (in Analysis section), states “...can be found in the supplementary material”. Similarly page 13, in Data analysis.), but I couldn’t find supplementary material, either at the end of the PDF, in the PeerJ review materials, nor in the OSF repository. This was not a big problem for me because all information is contained in the source code, but this should be fixed (if it was not my oversight).

Experimental design

no comment (everything is satisfying)

Validity of the findings

The statistical analysis methods are appropriate (e.g., using a full random-effects structures and an appropriately transformed dependent variable). While I did not check all the details in the source code, the code ensures reproducibility of reported results. Planned and exploratory analysis are cleanly separated. Conclusions are well stated and supported by the analysis results.

Minor comment (change encouraged but not mandatory):
The authors note that surprisal generally predicts that long dependencies reduce processing effort. Whether this actually applies to the verb-particle dependencies studied here is, however, not easy to establish due to data sparsity (as the authors explain convincingly). Intuitions about surprisal can be misleading, and simulations with a probabilistic model are needed to really understand what the predictions are. Given that the stated surprisal predictions are not supported by simulations, I suggest the authors temper their claim about the predictions of surprisal. This is already acknowledged in the "Predictions" section, but I'd suggest the authors also acknowledge it in the conclusion. E.g., in the conclusion: “the surprisal account would predict…” is an overstatement given that there are no simulation results or explicit claims about these particles in the cited surprisal studies to support this. After all, it might well be that the absence of evidence for a distance effect is exactly what surprisal would predict. The authors already acknowledge this possibility in the Conclusion, where they state that previous observed antilocality effects might have been due to stronger lexical constraints created by intervening material than in the stimuli here.
For what it’s worth, I ran five replicates of a high-quality neural language models (trained on 800M words of German Wikipedia text, about at the level of the models in Gulordava et al. 2018 (https://www.aclweb.org/anthology/N18-1108/)) on the stimuli to obtain surprisal estimates on the particle, predicting model surprisal from distance (long/short) and the entropy from the cloze study, with full random effects for items and model replicates. There was a clear effect of entropy (beta=1.2, t=3.1), at most marginal evidence for an effect of distance (beta=-0.17, t=-1.6), and no evidence for an interaction (t=-0.8). (Also, no evidence for any effects was observed when predicting from the discrete 2x2 contrasts of the experiment). So, while these surprisal models predict an effect of entropy, they predict at most a small effect of distance.
Given that there are no established neural language models for German that have been used in prior psycholinguistic research, there is no need at all to run such models for the purposes of this paper. However, these simulation results suggest being careful with evaluating surprisal predictions and emphasizing that the predictions described here are intuitive, not based on an actual probabilistic model. Alternatively, the authors could also acknowledge that it is simply not clear whether or not surprisal predicts an antilocality effect for these data.

---

## Round 0.2 · Minor Revisions

Dear Dr Stone,

I sent your revised manuscript to the two reviewers who suggested major changes in the first round of reviews. As you will see, one reviewer is satisfied with those changes while the second reviewer still requires a number of minor changes - most relating to improving the clarity of the manuscript. To maximise the impact of your work, it is important that your manuscript can be understood by as many people as possible - regardless of background - and hence I strongly suggest you consider all of the suggested changes of Reviewer 2 - to improve clarity. Once this has been done, I will revise the manuscript myself for readability, since it is quite a "dense" piece of work, and we want to make the content as accessible as possible. So, we may have a couple of revisions ahead of us, but if you are able to address all points of concern, I am hopeful for acceptance.

With best wishes,

Genevieve

·

Basic reporting

no comment

Experimental design

no comment

Validity of the findings

no comment

·

Basic reporting

no comment

Experimental design

no comment

Validity of the findings

no comment

Additional comments

I thank the authors for their serious and thorough consideration of my comments from the previous round of review. I find the manuscript much improved, clearer, more focused and easier to read and understand.
I still have some remaining relatively minor comments, mostly with regard to the presentation in the Introduction. If these concerns are addressed, the manuscript can be published in PeerJ.
Specific comments:
Abstract:
"Locality effects induced by interference and working memory have been…": this would probably be clearer if it said "induced by interference and working memory load have been…" (as in the previous sentence).
Also - and this is not crucial at all for the paper, I'm just wondering about it – I'm not sure what the authors mean by "effects of working memory load". The way I see it, interference can come about as an effect of high working memory load (more items that are similar to one another); decay can also come about as an effect of high working memory load (not enough resources to keep the item active while keeping other items active too). So I wonder if in saying "working memory load" the authors mean some other effect, possibly displacement, i.e. forgetting some material to make room for other material?

Introduction:
I think the presentation of surprisal and related ideas is still somewhat confusing, mainly because, I believe, there are two distinct, important predictions made by surprisal, but the presentation sort of mixes the two:
1. First, surprisal says "more predictable is easier". This is by no means something that was first claimed by surprisal theory. It was shown in ERP studies since Kutas & Hilliard (1980) and in reading times since Ehrlich & Rayner (1981) (and maybe before?). Surprisal is just one way to model this observation. This generalization is the basic tenet in the predictions of both theories contrasted in Figure 1 (in both, reading times on the left are shorter than on the right).
So I think in the subsection discussing word predictability, the discussion shouldn't really start with or focus exclusively on surprisal. In fact, on the next page the authors offer an explanation for the effect of predictability based on decay (lines 116-118). The discussion in the subsection discussing word predictability can therefore outline the main observation (predictable is easier) and findings, and then mention the surprisal account, and also the decay account.

2. The second thing, which is more specific to surprisal, is the prediction for antilocality effects. Antilocality is briefly explained in the abstract and then is sort of assumed, but never really presented methodically.
So I would suggest including an explanation of this effect in a dedicated subsection. So the order will be: the "word predictability" section (the prediction of which are identical for the two hypotheses); the "antilocality" section (suprisal); and then the "decay" section.
Also, the two paragraphs on the interaction of predictability with distance (p. 2 line 65 onward) are very confusing. They sort of go back and forth between discussing the interaction of predictability with distance and discussing the interaction of predictability with working memory load without explicitly explaining why or whether the two (distance/working memory load) are interchangeable.
Finally, the bottom line of these two paragraphs is "facilitation in the reading times of a distant word .. may only occur when that word is highly predictable" (this is also stated in the predictions section, and in the conclusion) – but this interaction is not represented in Figure 1, where the effect of distance is identical for more predictable and less predictable words.
One general suggestion: it would perhaps be helpful to have one example sentence in the introduction (perhaps even with a verb-particle dependency) to accompany the discussion, so the different predictions can be exemplified with regard to that sentence, to make them concrete and easier to understand.
Experiment 1 Methods:
I think it would be helpful to state explicitly, around line 307, that the set size manipulation therefore did not result in a difference in the predictability of the particle.
The entropy formula on line 320 should be explained.
One last thing – just a thought, no need to do this – I wonder whether in the eyetracking there would be effects on rates of skipping the particle altogether (since we know that more predictable words are skipped more often). The authors say that the particle was not always fixated – I wonder, for future studies, if there could be something interesting there.

Typos and very minor comments:
Abstract, 8th line from bottom: should be "decay, predictability or their interaction".
Abstract, 5th line from bottom: perhaps instead of "facilitate or hinder reading times", change to "facilitate or hinder processing"?
Line 210: parentheses missing around Lewis and Vasishth, 2005.

---

## Author Rebuttal · Round 0.2

Department of Linguistics
University of Potsdam
Karl-Liebknecht-Straße 24-25
14476 Potsdam, Germany

Prof Genevieve McArthur
Associate Editor
Dear Prof McArthur,

Thank you for your quick review even in view of the Australian summer break and for the extremely helpful and insightful comments. Please accept our sincerest apologies for the long delay in returning our revisions.

We felt that all three reviewers' suggestions served to improve the clarity and preciseness of the manuscript and have therefore incorporated all of their suggestions. The major comments related mainly to the Introduction section, finding that this section lacked sufficient detail on the two accounts under investigation. The majority of our changes have therefore been to the Introduction, which we hope you find improved.

In the response letter below, we present your and each reviewer's comments in bolded text, and provide our response under each comment. Note that in some cases, we have split the comments into smaller chunks and/or paraphrased the comment for brevity. Where line numbers are given in the bolded reviewer comments, these refer to the original manuscript. New page numbers referring to the revised manuscript are provided in our responses.

Thank you again for your time and valuable feedback in reviewing the manuscript and we hope that we have sufficiently addressed the concerns raised.

Yours sincerely,

Kate Stone
Dr. Titus von der Malsburg
Prof. Shravan Vasishth

# Responses to comments from the editor

1. **Predictions section (page 3). I found this section confusing. It seemed to "come out of the blue" - partly due to unclear wording, I believe, and partly because not much background had been provided about the two models that were being pitched against each other. I believe Reviewer 2 had a similar concern, and has offered some specific suggestions for how this might be addressed in the Introduction. In addition to those suggestions, please ensure that the Prediction section is precluded by a clear explanation of the two theories, and that the logic behind each prediction is described as clearly and simply as possible.**

   In order to address your concerns, we have used the suggestions of Reviewer 2 to restructure the Introduction. We hope you find that the revised Introduction now sets up the Predictions section more clearly. Specific examples of how the Introduction was revised are provided below under Reviewer 2's comments. Where possible, we provide specific page numbers where we have added requested information. However, the document containing tracked changes may, in some cases, be more useful where large sections of the Introduction have been reorganised.

2. **Participants section: Please clarify if "language" disorders include reading disorders.**

   We excluded participants with any kind of reading or production disorder and have rephrased this in the Participants sections on pages 5 and 12 to state "Participants were screened for acquired or developmental reading or language production disorders".

3. **Materials section: I was a bit confused by the presentation of the stimuli. Would it be possible to reformat the examples to improve clarity by adding a blank line between the two lines of the German/English stimuli, and also provide the meaning of the text prior to the stimuli?**

   Beginning on page 5, we have simplified this even further to condense each example into two sentences: one German and one translation. We have removed the English gloss altogether since we do not need

to annotate morpho-syntactic components. We provide one condition here as an example:

**Small set/short distance:**

Mit dem neu gekauften Lappen **schrubbte** sie die Teller in der Küche **ab**, um Platz zum Kochen zu schaffen.
*With the newly bought rag, she **scrubbed** the plates in the kitchen **off** to create space for cooking.*

4. **I understand why you might decide to outline the history of the development of the stimuli under Materials. However, the length of this history narrative the reader from the flow of information for Experiment.**

We agree that this section disrupts the flow of the main text. We have revised this section to present only the cloze test results in the main text on page 6, since they are the directly relevant to the experiments presented. The frequency analysis has been moved to the appendices and the norming study removed entirely.

5. **At some point, there appeared to be an abrupt switch from the use of the term "predictability" to "entropy". If they are the same thing, it would help the reader to use the term "predictability" throughout the manuscript, since it is a less specialised word. However, if a switch to entropy is required, this needs to be explained clearly at the appropriate point in the narrative.**

We appreciate that switching between terms without enough explanation has caused confusion. We do believe the switch is necessary, however, and have therefore revised the text to first state explicitly in the Introduction how predictability maps to set size (page 4), and that we also later be introducing the term entropy:

> "Note that throughout the remainder of the article, we use *set size* as a proxy for predictability. Set size also relates to *entropy*, which we introduce in detail as it becomes relevant in the Cloze Test section."

We then give a more detailed explanation of entropy where we first use it as a predictor in the cloze test analysis on page 7:

"The *set size* manipulation was intended to induce uncertainty about the upcoming particle's lexical identity. One useful way of quantifying uncertainty is with *entropy*. Entropy provides a measure of how much information is carried by a new input in light of all possible outcomes. In our case, the new input is the particle. In a sentence context where many particles are plausible and cloze probability is uniformly low across all the plausible particles, we assume that uncertainty about the identity of the upcoming particle is high. Thus, each of the plausible particles carries a large amount of information about the meaning of the sentence and entropy is high. In a sentence where only few particles are plausible and one particle is much more probable than the others, we assume that uncertainty about that particle's identity and the meaning of the sentence is low, and so encountering the high-probability particle will be less informative; this is a low entropy situation. To calculate entropy for our experimental stimuli, we first calculated the cloze probability (P) of all verb particles given for each respective sentence in the cloze test. Entropy (H) of the target particle was then defined as:

$$H = -\sum_i P_i log P_i$$

In the same section, we state explicitly how entropy maps to predictability:

"We therefore present an analysis of entropy as a continuous predictor instead, since this maps better to our planned manipulation of predictability (high entropy = low predictability and vice versa)."

Throughout the remainder of the paper, we attempt to use both terms where possible, e.g. "high entropy (low predictability)".

## Responses to comments from Reviewer 1

### Major comments

1. **Does the verb in every stimuli sentence require a particle? If not, how was the "no particle" option incorporated in the**

**cloze test and data analyses?**

All items were confirmed to require a particle via cloze testing and native speaker judgements. Any items that did not require a particle (as evidenced by the Cloze test) were presented to participants as fillers, but not included in the final analysis. The text has been updated to state this more explicitly, both in the Introduction on page 4: "Using a cloze test, we confirmed that each sentence required a particle."; and under Materials on page 5: "Each experimental item was a quartet of four sentences in which the context required a particle for the sentence to be grammatical.".

2. **There is no mention of how spillover was taken into account, even though this phenomenon is prevalent in reading, in particular self-paced reading. Were reading times on words directly following the particle also taken considered? If not, could this be why the expected effects were not found?**

We did originally consider looking at the spillover region, but decided against is because the particle must be followed by a comma and it was not clear how the clause boundary and associated sentence wrap-up effects (Rayner et al., 2000) might interact with reading times in the spillover region. We therefore presented mean reading times across the sentence in Figure 1 (below). Figure 1 does not suggest that there was any difference in reading times in the spillover regions other than in the long-distance eye-tracking data where we already saw effects at the particle, and thus we did not analyse or discuss the spillover region further.

We have updated the text in both Data Analysis sections to include a statement about the spillover region:

> "We also considered analysing the spillover region, but decided against it as the particle had to be followed by a comma and it was not clear how the clause boundary and associated sentence wrap-up effects (Rayner et al., 2000) might interact with reading times in the spillover region. Instead, we present mean reading times across the sentence in Figure 1."

[Figure]

Figure 1: **Comparison of self-paced reading times and eye tracking total fixation times plotted across the sentence.** Error bars show 95% confidence intervals.

## Minor comments

- **Line 124: what does it mean for something to be "anecdotally assumed"?**

  Here we were trying to express the fact that some papers assume acti-

vation decay has played a role in their findings on long-distance dependencies even though they don't specifically test it. We have updated the sentence on 3 to say this more explicitly:

> "There are few empirical experiments specifically testing decay in isolation, even though it is generally presumed to affect word processing times in long-distance dependencies (e.g. Xiang et al., 2014; Ness and Meltzer-Asscher, 2019; Chow and Zhou, 2019)."

- **When introducing German particle verbs, it would be good to mention that moving the particle to after the object NP is required in German.**

A sentence has been added to the Introduction to make this explicit on 3:

> "In German, however, the particle must appear after the direct object if the verb is transitive, usually at the right clause boundary (e.g. "Er raümte den Raum auf" *he tidied the room up*, but not "*Er raümte auf den Raum" *he tidied up the room*; Müller, 2002)."

- **Line 115: the Dutch prefix "ver" in "verdelen" is not a particle (i.e., it is not split: "hij deelt het ver" is not possible)**

This example has been replaced with an example from (Piai et al., 2013) on page 4:

> "In this study, it was hypothesised that Dutch verbs that can take a large number of possible particles (e.g. *spannen*, "to tense", which can take at least seven particles) would trigger preactivation of those particles, placing a larger demand on working memory than verbs with a small set size (e.g. *kleuren*, "to colour", which can take only two)."

- **Line 127-128: "self-paced reading and eye tracking modalities" and "reading modalities": shouldn't this be "paradigm" instead of "modality"? In both cases, the modality is written/visual.**

The word "modality" has been replaced with "experimental method" throughout the manuscript. We opted against "paradigm", only be-

cause we use this word elsewhere to describe the experimental presentation method.

- **Table 1 shows 95% CI instead of the standard error mentioned on line 221. Also, the caption is not quite accurate because the table presents cloze statistics but not the cloze test results.**

The text on page 6 has been amended to state "Means and 95% confidence intervals of Beta distributions corresponding to the cloze probabilities for each factor level are presented in Table 1." and the caption of this table has been updated to state "cloze statistics" instead of "cloze test results".

- **It would be helpful if the goal of the cloze test data analysis were explained before the technical details.**

We have revised the beginning of the cloze test section on page 6 to state the three purposes of the cloze test:

> "In order to confirm that our sentence stimuli (i) elicited particles, (ii) that more particles were elicited by the large set condition than the small set condition, and to (iii) quantify the predictability of the target particle, a cloze test was conducted."

- **Cloze test analysis results: "the probability of the target particle was lower ... for the interaction": For which combination of factor levels was the probability lower?**

The posterior for the interaction effect suggests there was no interaction of set size or distance that predicted cloze probability. Looking at the nested effects, there is a very small difference in cloze probability between large/small set verbs at long distance (lower probability for large set verbs), whereas there is no such difference at short distance. However, the difference in cloze probability between set sizes at long distance is about 0.025%, so this is not really meaningful. We have revised this paragraph on page 7 to remove any mention of the probability being lower and simply state that the posteriors are consistent with zero:

"The model did not suggest that either set size, distance, or an interaction of the two influenced cloze probability. As can be seen in Figure 2, the posteriors for the probability of giving the target particle were more or less centred on zero, meaning that neither set size, distance, or their interaction made people any more or less likely to give the target particle."

- **The violin plots of Fig. 3 shows probability mass for negative values of entropy, even though entropy is by definition non-negative.**

This was an error in the plot code and has been amended. The amended plot can be found on page 8.

- **Line 414: what did the preprocessing of eye-tracking data entail?**

Apologies for having omitted this from the manuscript. The following detail has been added to the Analysis section of the eye-tracking section on page 15:

"Sampled data were exported from DataViewer (SR Research) and pre-processed in R using the *em2* package (Logačev and Vasishth, 2013). Trials containing blinks or track loss were excluded."

- **Line 417: the citation to R is "R Core Team", not just "Team".**

This has been amended throughout the manuscript.

- **Line 448-449: the problem of evaluating multiple dependent measures is not a "limitation of the BF analysis" in particular, is it?**

This is a good point, thank you. It is not a limitation of BFs, but rather of the way we have used them: applied to multiple dependent measures with no FWER correction. While the reference we cite relates to FWER in a frequentist framework, there may be an analogous issue in Bayesian analyses, but there is no current option for controlling or

correcting for this. We wanted to highlight this point and the fact that further confirmatory analysis is needed. We have rephrased this sentence on page 16 accordingly to state:

> "A possible limitation of our approach using Bayes factor analyses is that we are evaluating multiple measures, without any correction for family-wise error (**?**). While the family-wise error rate is a frequentist concept, it may be that an analogous issue exists in the Bayesian framework for which we have not controlled. Our analyses should therefore be considered exploratory and confirmed via future replication attempts.".

- **line 474: "The statistical analysis" should probably be "The outcome of the statistical analysis"**

This text no longer appears in the manuscript.

## Responses to comments from Reviewer 2

### Major comments

- **Clarity of opposing hypotheses, particularly the predictions of the LV05 model: If I understand correctly, the experiments set out to test the predictions of Surprisal vs. LV05. What is Surprisal theory, what are its main tenets? What is the LV05 model? What is it modelling, what are its assumptions?**

Our intent was actually to test surprisal versus decay rather than versus LV05 specifically, but we agree that the way the Introduction was formulated created confusion. We have therefore reframed the Introduction such that we compare the opposing predictions of predictability (as instantiated by surprisal) and temporal activation decay.

The Introduction first introduces predictability, beginning with this paragraph on page 2:

> "**Word predictability.** The surprisal theory of sentence processing provides an account of how words in a sentence become predictable and how predictability facilitates their processing (Levy, 2008; Hale, 2001). Surprisal is based on the assumption that the context of a sentence sets up expectations about what structural

information might appear next. Under surprisal, the difficulty of processing each new word in a sentence is equal to the negative log probability of that word appearing given the preceding context. The probability of a word given a context can be quantified using a probabilistic context-free grammar (PCFG; e.g. Levy, 2008). At each new word in a sentence, a set of plausible sentence continuations is generated based on the PCFG and held in parallel, ranked by their frequency. The degree of update that each new word induces in the distribution of probabilities over these structures is proportional to the difficulty of processing the new word; that is, the greater the update, the greater the processing difficulty or "surprisal". In broader terms, this means the more constraining a sentence is, the fewer likely possible continuations it will have and therefore the lower surprisal will be at an expected word. Conversely, at an unexpected word, surprisal will be higher. Lexical constraints are often not explicitly modelled in surprisal (Levy, 2008; Hale, 2001), but lexicalised PCFGs have demonstrated that their contribution to processing difficulty follows a similar pattern (Collins, 2003; Charniak, 2001)."

The Introduction then introduces decay, beginning with the following paragraphs on page 2:

"**Temporal activation decay.** A less well-studied factor in dependency processing is temporal activation decay. Decay is assumed to affect sentence processing in the following way: At any new word in a sentence, there may be a number of ways the sentence structure could plausibly continue. For example, the sentence *The secretary forgot...* could continue with a direct object NP (e.g. *the files*) or with a clause (e.g. *that the student...*); it has been proposed that both of these structures may be activated, but that only one will be pursued by the parser while the other is left to decay (Van Dyke and Lewis, 2003). Thus, if the parser pursues the sentence structure assuming an upcoming NP, but instead encounters the word *that...*, the decayed structure must be reactivated and reading time at the word *that* will be slower than if the expected NP had been encountered (Ferreira and Henderson, 1991; Gibson, 1998; Van Dyke and Lewis, 2003). Even if the NP parse proves to be correct, activation of the NP will decay over time such that, if it must be retrieved later (e.g. as the antecedent of a relative clause), retrieval time will become slower if

the retrieval is delayed (Lewis and Vasishth, 2005).

The above example concerns structural continuations of the sentence, but plausible continuations may also include the preactivation of specific lexical items, with the most probable item pre-integrated into the building sentence parse if its activation is strong enough (Kuperberg and Jaeger, 2016; Ness and Meltzer-Asscher, 2018). As for the structural example above, it can be assumed that lexical items preactivated but not pre-integrated are left to decay. Likewise, if future input indicates that the wrong lexical item was pre-integrated, then the decayed, correct item can be reactivated in order to repair the sentence, reflected by longer reading times. Reading times should therefore be faster if there is only one, highly probable lexical item, because the probability that the parser pursues a parse with the wrong lexical item will be low. With an increasing number of plausible lexical items, reading times should be slower, because the probability that the parser pursues a parse with the wrong lexical item increases and the reactivation of decayed items will occur more often. Even if the correct lexical item is pre-integrated, this item may too be subject to decay. However, due to stronger preactivation from the context, more predictable items are likely to have a higher starting activation and thus the effects of decay will not be as severe. Under these assumptions, less predictable lexical items are, on average, more sensitive to the effects of decay than more predictable items, leading to a more pronounced reading time slow-down (a locality effect) at less predictable dependency resolutions."

We no longer make reference to the LV05 model in the Introduction since the focus of that model is interference, which is not relevant to the current experiments. We had originally thought that LV05 might be a good framework for describing the predictive process for our particle verb stimuli and how this might interact with decay. LV05 does contain an element of predictive processing, in that it anticipates upcoming structure, and an explicit decay parameter. However, for the revision, we decided against this approach as it raised more questions than it answered. We do use the decay parameter of LV05 to simulate the effect of decay, but limit our discussion of this to the Predictions section.

- **Then, the authors should explain both frameworks' hypothe-**

**ses about decay and its interaction with predictability. More explanation is needed, along with the relevant results (from German? Hindi? Persian?).**

The Introduction has been revised to include a more specific account of how decay and surprisal might interact with lexical predictability; see the response to Reviewer 2's first question above. Here the changes are substantial and we refer reviewers to the sub-sections "Word predictability" and "Temporal activation decay" within the revised Introduction. In the second and third paragraphs under "Word predictability", we review evidence for the interaction of predictability with surprisal, specifically the finding that surprisal may only be a good predictor of reading times in high predictability sentences with low working memory load (Levy and Keller, 2013; Husain et al., 2014); although we note that this finding has been difficult to replicate (Vasishth et al., 2018). Then, under "Temporal activation decay", we describe a mechanism for how predictability might interact with decay by assuming that the same process of decay for structural material (Ferreira and Henderson, 1991; Gibson, 1998; Van Dyke and Lewis, 2003; Lewis and Vasishth, 2005) also applies to lexically preactivated material (Kuperberg and Jaeger, 2016).

- **Line 149 onwards "in the absence of interference, decay over distance ... will make the long condition more sensitive to predictability". Why? Do the authors claim that when a lexical item is highly predictable, it is integrated (prior to its occurrence in the input) and it is therefore amenable to decay? If so, it should be stated clearly.**

We have revised the section on page 2 to explain this more clearly. The revision of this section is large and so we do not quote it here, but to summarise: we assume that the effects of decay will show up more in *less* predictable items. The reason for this is that accounts of serial parsing propose that multiple plausible structural continuations of a sentences may be activated, but that only one parse structure is pursued while the others are left to decay (Ferreira and Henderson, 1991; Van Dyke and Lewis, 2003; Gibson, 1998). We then make the assumption that upcoming lexical items can also be pre-integrated into the pursued parse, especially if their identity is certain enough (Ku-

perberg and Jaeger, 2016; Ness and Meltzer-Asscher, 2018). However, when there is more uncertainty, the chance that the wrong word is pre-integrated would increase and the correct word will be left to decay. Thus when the real word is encountered, reactivation of the decayed, correct word will be necessary, increasing reading time. It may be that even when uncertainty is high, the correct word is still pre-integrated, but on average, this probability should be lower in the low predictability/high entropy condition. In contrast, a correct pre-integrated word (of which the probability is higher in the high predictability/low entropy condition) will not decay. Some accounts propose that decay only affects the structure pursued in working memory (Lewis and Vasishth, 2005), in which case the pre-integrated particle itself may also be subject to decay. Here we assume that more predictable particles will have a stronger starting activation, so the effects of decay at the particle site will not be as pronounced as for less predictable particles with a lower starting activation.

- **What's "highly predictable"? Consider for example a verb from the small set size group which takes five possible particles. If one of them appears in 80% of cases, and each of the other four – in 5% of cases, is the most probable one highly predictable, therefore integrated and amenable to decay? What about a "small set" verb with 60%-10%-10%-10%-10% distribution of particles and a "large set" verb with 60%-4%-4%... distribution? What happens when there's no one highly predictable completion?**

This is certainly an important point and one that we feel is covered by the exploratory analysis using entropy instead of set size. Our quantification of entropy takes into account the distribution of possible particles activated and indeed indicated that there were experimental sentences in the small set condition (supposedly low entropy/high predictability) that actually had high entropy (low predictability) values, and vice versa for the large set condition. By collapsing the small/large set categories and using the entropy values as a predictor instead, this should mean that the distribution of particles at the low entropy (high predictability) end of the scale would have a distribution more like 60%-4%-4% and at the high entropy (low predictability) end of the scales, more like 30%-30%-30%.

It is true that we generally assume the highest probability word is integrated, even if the most probable word is only slightly more likely than other plausible words. For example, the distribution of probabilities may be something like 35%-30%-30%). In this case, the most probable word would still presumably be integrated. We would not consider this a "high predictability" situation however, because the small difference in probability might make it more likely that noise results in one of the other words being pre-integrated instead (e.g. the noise parameter in LV05 means that sometimes a word other than the highest activated word is (mis)retrieved). If there were no one highly predictable completion, then which particle gets pre-integrated could be random. It could also be that no particle is pre-integrated, depending on what the activation threshold for pre-integration is (based on your model in (Ness and Meltzer-Asscher, 2018)). The preactivated-but-not-pre-integrated particles could therefore still be affected by decay (the sentences required a particle so we assume that *something* was preactivated). In any case, all of these scenarios are captured by the continuous variable *entropy*.

- **The upshot from the last two questions is: shouldn't we look at constraint (cloze probabilities) \*at the verb\* in order to know what was preactivated/integrated there? Or perhaps at entropy, if it is assumed to modulate preactivation/integration (e.g. integration only happens when there are no strong competitors, i.e. low entropy), but again, \*at the verb\*?**

The degree of constraint at the verb is definitely critical; however, to measure particle preactivation at the verb with a cloze test would be difficult. Because the context is so unconstrained at the verb region of the sentence, non-particle completions would represent a high number of cloze completions and we would need a large amount of data to get non-zero frequency counts for each particle, especially for verbs that take 10s of particles. Thus, while we assume particle preactivation occurs at the verb, it may only become strong enough to be detectable later in the sentence when the verb is combined with its arguments – at what exact point detectable preactivation occurs, it is difficult to know. For this particular paper, we therefore focused on whether preactivation could be sustained rather than on when it was triggered.

- The manuscript does discuss entropy, but measured right before the particle. In the pre-test, it turns out that there's no difference between the two groups, but this is only discussed in the Results section, before carrying out the alternative analysis. I think it would be much better to acknowledge the potential problem when the pretest is presented.

Thank you for the recommendation. We have amended the pre-test section on page 7 by moving the discussion of the entropy issue here from the results section. We also state that although we will still present the planned analysis for transparency, the exploratory analysis with entropy is more relevant.

- How were the verbs selected? Based on the cloze pretest, namely based on their preference after the object, before the particle? Or based on their particle selection options regardless of the specific object?

A section has been added to the materials section on 5 to explain how the stimuli were created:

> "To develop the experimental stimuli, verbs were first selected
>
> using a corpus and dictionary search of verbs and all their possible particles. Verbs and their particle sets were grouped into small (fewer than 6 particles) and large (greater than 10 particles) categories and sentences constructed by German native speakers around small/large set pairings."

and to the cloze text section on 6:

> "An initial total of 48 items, each with 4 conditions (a-d), was truncated just before the particle such that the verb and the direct object of the sentence were known."

## Other comments

- I think it is natural to start the Introduction with the discussion of decay (which now appears in the second paragraph), as these are the more traditional approaches to distance effects. Then, Surprisal and anti-locality can be presented.

This suggestion has been included in the restructure of the Introduction. Because the restructure of the Introduction is substantial, specific page references are difficult to present here. The tracked changes document may be more useful in reviewing the changes made.

- **The manipulation of decay was introduced by adding a very short constituent – a two-word phrase. Could that be the reason why no effect of decay was found? Does the LV05 predict an effect of decay with such a minimal manipulation? Related to this, line 526, "it would have been difficult to construct longer sentences without reintroducing these factors (interference), which supports the idea that they are the source of processing difficulty": why does it support this idea? I think it only means that it's very hard (perhaps impossible?) to test the influence of decay by itself.**

Yes the constituent is very short – the example item contains a particularly short intervener, but others were longer (although not by much). The results of our simulations with the decay parameter of LV05 (see Figure 2, right panel, below) do predict a small amount of decay for a short constituent, but it is definitely possible that decay was undetectable in our stimuli.

[Figure]

Figure 2:   **Predicted interaction of lexical predictability and distance.** Informal predictions of the surprisal account and a simulation using the decay parameter of the LV05 model.

However, the fact that decay is hard to test in isolation without introducing interference as a confound suggests that decay may just be

outweighed by interference in terms of its contribution to processing difficulty. This has certainly been the conclusion of a number of studies: we have expanded the discussion on this issue under Temporal Decay on page 21:

> "The evidence against an effect of temporal decay in both self-paced reading or eye tracking is entirely consistent with findings suggesting that decay is not an important factor influencing reading and memory recall times (Lewandowsky et al., 2009; Engelmann et al., 2019; Vasishth et al., 2019). In comparison to the sentences used in previous research, the sentences used in the current study were relatively simple, without interference or a particularly high working memory load added by the distance manipulation. However, the short adjectival modifiers used to introduce decay in our experimental stimuli may not have been long enough to introduce a detectable effect of decay. It would have been difficult to construct longer interveners without reintroducing interference or working memory load, which could support the idea that interference and working memory load are indeed the source of processing difficulty in longer sentences, rather than temporal decay. Alternatively, it could be argued that the difficulty in constructing longer sentences without introducing interference or working memory load means it is difficult or impossible to test decay in isolation and thus that we cannot know what the true effect of decay is. However, if the effect of decay is so small that it is undetectable in the face of interference and working memory load, and that these factors are almost unavoidable in constructing long dependencies, then decay is, as mentioned above, likely not a major influence on processing difficulty."

- **When entropy is first discussed, the concept should be explained – not only with a mathematical formula, but also with the intuition as to what it means.**

A more detailed explanation has been added to the Cloze Test section on page 7:

> "The *set size* manipulation was intended to induce uncertainty about the upcoming particle's lexical identity. One useful way of quantifying uncertainty is with *entropy*. Entropy provides a measure of how much information is carried by a new input in light

of all possible outcomes. In our case, the new input is the particle. In a sentence context where many particles are plausible and cloze probability is uniformly low across all the plausible particles, we assume that uncertainty about the identity of the upcoming particle is high. Thus, each of the plausible particles carries a large amount of information about the meaning of the sentence and entropy is high. In a sentence where only few particles are plausible and one particle is much more probable than the others, we assume that uncertainty about that particle's identity and the meaning of the sentence is low, and so encountering the high-probability particle will be less informative; this is a low entropy situation. To calculate entropy for our experimental stimuli, we first calculated the cloze probability (P) of all verb particles given for each respective sentence in the cloze test. Entropy (H) of the target particle was then defined as:

$$H = -\sum_i P_i log P_i$$

## Minor comments

- **Line 47 "activation decay is anecdotally assumed...": another relevant reference here is Chow & Zhou (2019), which is a replication of Wagers & Phillips (2014) (though the original authors do not frame their study as investigating decay).**

  Thanks! We have included it on line 130.

- **Line 52 "decay is not a useful predictor": perhaps also cite Van Dyke and Johns' (2012) review which argues against a role for decay in sentence processing.**

  This reference has also been included, on line 124, thank you for mentioning it.

- **Materials section: Do all the experimental verbs necessarily take particles at all? I assume this is the case, but I think this should be stated explicitly.**

All items were confirmed to require a particle via cloze testing and native speaker judgements. Any items that did not require a particle (as evidenced by the Cloze test) were presented to participants, but not included in the final analysis. The text has been updated to state this more explicitly, both in the Introduction on page 4: "Using a cloze test, we confirmed that each sentence required a particle."; and under Materials on page 5: "Each experimental item was a quartet of four sentences in which the context required a particle for the sentence to be grammatical.".

- **Line 217 "24 items that suited the experimental design" – meaning what? That they selected 6 or less, or 15 or more, particles?**

Exactly: not only did the items elicit the required number of particles (less than 6 or more than 10), but they always elicited a particle. Detail has been added to the beginning of the Cloze test section on page 6 to state specifically how items were selected:

> "In order to confirm that our sentence stimuli (i) elicited particles, (ii) that more particles were elicited by the large set condition than the small set condition, and to (iii) quantify the predictability of the target particle, a cloze test was conducted."

- **Online norming study (line 249 onwards): Why is this pretest necessary? In the experiment, the verb is several words upstream from the particle, so why are reading times of the verb+particle relevant?**

It was felt that there might be some property of verb-particle constructions that leads to them being read faster or slower depending on the number of particles they take even when the verb and particle aren't separated. Perhaps, for example, base verbs that take more particles have more lexical and/or semantic associations with other lexical entries and thus net activation via passive spreading might be lower than for a verb that has fewer associates. Since we were interested in whether distance was the key factor in reading time changes, we needed to rule this possibility out. However, we have now removed this section since it is tangential to the main text.

- **Line 382, "a second possibility is that locality and antilocality**

**effects simply cancelled each other out": how is this relevant to the effect of predictability, which is the topic of discussion? I would think that it is relevant to the (lack of) effect of decay, not predictability.**

The suggestion that locality and antilocality may have cancelled each other out was a point about the mathematical consequences of averaging speed-ups and slow-downs, rather than a theoretical point related to predictability. However, this text no longer appears in the revised manuscript as we do not investigate it further due to space considerations.

- **Line 484 "speed up at the verb": this sounded to me like the authors were referring to a speed up at the verb relative to preceding material; it took me some time to understand that it means lower reading times in the large set verbs compared to the small set verbs.**

This section on page 10 was indeed about the base verb region of the sentence – we do not conduct any analysis with the base verbs as they are not matched, but mention them here because it was odd that reading times were not faster for large set/high entropy verbs, despite having higher corpus frequency. We have rephrased this section to make this clearer:

> "Mean reading times across the whole sentence for both experiments are plotted in Figure 1. One feature of these data that should be mentioned is that base verbs for sentences with higher entropy at the particle site had a higher corpus frequency than base verbs in sentences with lower entropy at the particle site (to compare verb frequency, we divided sentences into high and low entropy categories via a median split; see Appendix 2). Higher corpus frequency of the base verb should have resulted in faster reading times at the verb in high entropy sentences (Kliegl et al., 2004; Rayner and Duffy, 1986), but this was not the case in either experiment. The lack of a frequency effect at the base verb is discussed in the *General Discussion*."

- **Line 544, "a potential explanation for the lack of speed-up... more preactivated particles may have led to slower reading".**

**I'm not sure I would predict this. I would think activations are not usually viewed as costly. Perhaps the source of increased reading times here is that the verbs are more ambiguous/vague, i.e. have more possible meanings?**

This is a really good point and precisely what we were trying to get at – it is possible that the higher entropy base verbs themselves are more ambiguous, but a big driver of this ambiguity is the fact that they can be combined with a larger number of particles which change their meaning: so because readers can't see the particle immediately, the meaning of the verb is initially more ambiguous. This ambiguity, we assume, is associated with the increased range of preactivated particles, so I think it is conceivable that a larger amount of preactivated lexical material either results from or creates ambiguity and could be costly – this would be the idea behind the slower reading of low-constraint sentences, for example. We have updated the text to make our hypothesised link between preactivations and cost more explicit:

> "A potential explanation for the lack of a speed-up is that lexical entropy at the particle site reflected preactivation of particles at the verb. More preactivated particles would make the meaning of the verb more ambiguous, which in turn may have led to slower reading and cancelling out of the expected speed-up associated with higher frequency."

## Typographical errors

The following typographical errors have been amended:

- **Line 34: length should be amount** (this sentence no longer appears in the revised Introduction)

- **Line 166: items should be item**

- **Line 128: delete second 'also'**

- **Line 319: delete second 'the'**

Other comments:

- **Line 341 and caption for Figure 5: I initially thought the RTs in the table are reading times for the particle (and wondered why they were so high). The text and caption should say that these are RTs for answering the comprehension questions. Same for line 440 and table 9.**

  The relevant text and figure captions on pages 9 and 15 have been updated to explicitly state "question response accuracy and reaction times".

- **Line 389: the number "1" is missing.**

  Here we have spelt out "one" as per APA guidelines.

- **Line 457, "the results of the statistical analysis": in all the reading time measures? If so, maybe "analyses"?**

  This has been updated to "analyses" on page 20.

## Responses to comments from Reviewer 3

### Major comments

- **Supplemental material is referenced... but I couldn't find supplementary material, either at the end of the PDF, in the PeerJ review materials, nor in the OSF repository.**

  This may be a lack of clarity on our part - the phrasing in the original text could be interpreted to mean that the code and the supplementary materials were separate entities. What Reviewer 3 found on OSF was indeed the entirety of the supplementary material. Throughout the manuscript, we have therefore updated any reference to code to simply state "code" rather than "code in the supplementary materials".

### Minor comment (change encouraged but not mandatory):

- **Given that the stated surprisal predictions are not supported by simulations, I suggest the authors temper their claim about the predictions of surprisal [or]..., alternatively, the authors could also acknowledge that it is simply not clear**

**whether or not surprisal predicts an antilocality effect for these data.**

This is absolutely correct and we have tempered any statements about the predictions of the surprisal model to reinforce that they were informal; for example, in the Predictions section on page 4, where we have stated:

> "In the absence of formal quantifications for whether surprisal would predict an antilocality effect for our sentences, these predictions should be taken as an approximation of surprisal's general claim that long distance should always result in faster reading times and that higher lexical predictability should further sharpen expectations (Levy, 2008).";

as well as in the Conclusions section on page 22, where we compare our results to our predictions:

> "We compared two hypotheses of dependency processing in separable verb-particle constructions: informal predictions based on the surprisal account suggested that delaying the appearance of a verb particle could have elicited an antilocality effect, stronger in high vs. low predictable particles (Levy, 2008);"

As an aside: we would like to thank Reviewer 3 very much for the simulation results, these are very reassuring and even seem to resemble the pattern of reading times we found; at least in eye-tracking.

# References

Charniak, E. (2001). Immediate-head parsing for language models. In *Proceedings of the 39th Annual Meeting on Association for Computational Linguistics*, ACL '01, pages 124–131, Toulouse, France. Association for Computational Linguistics.

Chow, W.-Y. and Zhou, Y. (2019). Eye-tracking evidence for active gap-filling regardless of dependency length. *Quarterly Journal of Experimental Psychology*, 72(6):1297–1307. Publisher: SAGE Publications.

Collins, M. (2003). Head-Driven Statistical Models for Natural Language Parsing. *Computational Linguistics*, 29(4):589–637. Publisher: MIT Press.

Engelmann, F., Jäger, L. A., and Vasishth, S. (2019). The effect of prominence and cue association on retrieval processes: A computational account. *Cognitive Science*, 43(12).

Ferreira, F. and Henderson, J. M. (1991). Recovery from misanalyses of garden-path sentences. *Journal of Memory and Language*, 30(6):725–745.

Gibson, E. (1998). Linguistic complexity: locality of syntactic dependencies. *Cognition*, 68(1):1–76. ISBN: 0010-0277.

Hale, J. (2001). A probabilistic earley parser as a psycholinguistic model. *NAACL '01: Second meeting of the North American Chapter of the Association for Computational Linguistics on Language technologies 2001*, pages 1–8.

Husain, S., Vasishth, S., and Srinivasan, N. (2014). Strong expectations cancel locality effects: Evidence from Hindi. *PloS one*, 9(7):e100986. Publisher: Public Library of Science.

Kliegl, R., Grabner, E., Rolfs, M., and Engbert, R. (2004). Length, frequency, and predictability effects of words on eye movements in reading. *European Journal of Cognitive Psychology*, 16(1/2):262–284. ISBN: 0954-1446.

Kuperberg, G. and Jaeger, T. F. (2016). What do we mean by prediction in language comprehension? *Language Cognition & Neuroscience*, 31(1). ISBN: 2327-3798 2327-3801.

Levy, R. (2008). Expectation-based syntactic comprehension. *Cognition*, 106(3):1126–1177.

Levy, R. and Keller, F. (2013). Expectation and locality effects in German verb-final structures. *Journal of Memory and Language*, 68(2):199–222. Publisher: Elsevier Inc.

Lewandowsky, S., Oberauer, K., and Brown, G. D. A. (2009). No temporal decay in verbal short-term memory. *Trends in Cognitive Sciences*, 13(3):120–126.

Lewis, R. L. and Vasishth, S. (2005). An activation-based model of sentence processing as skilled memory retrieval. *Cognitive science*, 29(3):375–419.

Logačev, P. and Vasishth, S. (2013). em2: A package for computing reading time measures for psycholinguistics.

Müller, S. (2002). Particle Verbs. In Müller, S., editor, *Complex predicates: verbal complexes, resultative constructions and particle verbs in German.*, pages 253–390. CSLI: Leland Stanford Junior University.

Ness, T. and Meltzer-Asscher, A. (2018). Predictive Pre-updating and Working Memory Capacity: Evidence from Event-related Potentials. *Journal of Cognitive Neuroscience*, 30(12):1916–1938.

Ness, T. and Meltzer-Asscher, A. (2019). When is the verb a potential gap site? The influence of filler maintenance on the active search for a gap. *Language, Cognition and Neuroscience*, 34(7):936–948.

Piai, V., Meyer, L., Schreuder, R., and Bastiaansen, M. C. M. (2013). Sit down and read on: Working memory and long-term memory in particle-verb processing. *Brain and Language*, 127(2):296–306.

Rayner, K. and Duffy, S. A. (1986). Lexical complexity and fixation times in reading: Effects of word frequency, verb complexity, and lexical ambiguity. *Memory & Cognition*, 14(3):191–201.

Rayner, K., Kambe, G., and Duffy, S. A. (2000). The effect of clause wrap-up on eye movements during reading. *The Quarterly Journal of Experimental Psychology Section A*, 53(4):1061–1080. Publisher: Routledge _eprint: https://doi.org/10.1080/713755934.

Van Dyke, J. A. and Lewis, R. L. (2003). Distinguishing effects of structure and decay on attachment and repair: A cue-based parsing account of

recovery from misanalyzed ambiguities. *Journal of Memory and Language*, 49(3):285–316. Publisher: Elsevier.

Vasishth, S., Mertzen, D., Jäger, L. A., and Gelman, A. (2018). The statistical significance filter leads to overoptimistic expectations of replicability. *Journal of Memory and Language*, 103:151–175.

Vasishth, S., Nicenboim, B., Engelmann, F., and Burchert, F. (2019). Computational models of retrieval processes in sentence processing. *Trends in Cognitive Sciences*.

Xiang, M., Dillon, B., Wagers, M., Liu, F., and Guo, T. (2014). Processing covert dependencies: an SAT study on Mandarin wh-in-situ questions. *Journal of East Asian Linguistics*, 23(2):207–232.

---

## Round 0.3 · Minor Revisions

Dear Kate and fellow authors,

Thank you for putting so much effort into revising your manuscript in line with reviewers' last set of suggestions. These changes were rather extensive, and so I asked Reviewer 2 to have another look at your manuscript to make sure they addressed her suggestions. She was very happy with your work.

I have proof-read the manuscript - as a non-expert in the field - and have identified a few minor issues with readability that I outline below. All easy to fix.

1. There are a few sentences that require some minor adjustments in wording to fix the grammar. I have highlighted these sentences in the attached PDF.

2. Throughout the manuscript, there is inconsistent use of hyphen between particle and verb (e.g., particle verb and particle-verb and vice versa). Choose one convention and use consistently.

3. I am pretty sure PeerJ will want references in parentheses in alphabetical order. Please go through and amend throughout the manuscript. Also, please double-check the formatting for references in the text, and make sure you use PeerJ conventions in terms of the use of "and" or "&" etc. [** 4. The use of the term "surprisal" is grammatically correct in English. Surprisal account is perfectly OK. I suggest you identify all sentences that include the word "surprisal" on its own, and revise them to accommodate "surprisal account" in a grammatically appropriate way.

5. Your manuscript is very "dense" in terms of terminology. Please avoid the use of acronyms in the text (e.g., NP, BF) because it makes it just that much harder for the reader to follow your meaning.

6. I am not sure why the distributions throughout the manuscript are called "posteriors". I suggest you just called them distributions (google posteriors and you will see why - actually, don't!! do that - just look up the meaning of posteriors).

7. Table 8 and similar. Please provide full version of acronyms either in table (you may have room in column 1) or in the title or the notes.

8. When putting things in lists, make sure you used the PeerJ formatting for numbers (e.g., i, ii, iii OR (1), (2), (3) etc).

9. If a number is less than 10, then write in full (e.g., nine).

Once you have made these changes, I will forward in the PeerJ process towards proof editing.

Congratulations on all your hard work, and I look forward to seeing this in press.

Best wishes,

Genevieve

·

Basic reporting

NA

Experimental design

NA

Validity of the findings

NA

Additional comments

I thank the authors for the very thorough revision of the Introduction! I find the Introduction extremely clear now. The discussion around example (1) in particular is very helpful and accessible (and I believe it will benefit other readers too). I'm looking forward to teaching this paper in my classes!

One tiny comment, on line 119: please clarify what PCFGs stands for.

---

## Author Rebuttal · Round 0.3

Department of Linguistics
University of Potsdam
Karl-Liebknecht-Straße 24-25
14476 Potsdam, Germany

Prof Genevieve McArthur
Associate Editor
Dear Prof McArthur,

Thank you for your response regarding our manuscript and please accept my apologies again for the delay in responding (our lab hosted two conferences and one summer school over the the last 3 weeks—between that and the pandemic, everything has been somewhat chaotic).

We found Reviewer 2's suggestions very helpful, and have used them to reorganise the Introduction. We address the reviewer's comments point by point below. These represent the major changes to the manuscript, but other minor phrasing changes have been made throughout to improve clarity. The content/results are otherwise unchanged.

Once again we are very appreciative of your feedback and hope that you find the revised version improved.

Yours sincerely,

Kate Stone
Dr. Titus von der Malsburg
Prof. Shravan Vasishth

# Responses to comments from Reviewer 2

## Abstract

- **"Locality effects induced by interference and working memory have been...": this would probably be clearer if it said "induced by interference and working memory load have been..." (as in the previous sentence).**

  The missing "load" has been inserted.

- **Also - and this is not crucial at all for the paper, I'm just wondering about it – I'm not sure what the authors mean by "effects of working memory load". The way I see it, interference can come about as an effect of high working memory load (more items that are similar to one another); decay can also come about as an effect of high working memory load (not enough resources to keep the item active while keeping other items active too). So I wonder if in saying "working memory load" the authors mean some other effect, possibly displacement, i.e. forgetting some material to make room for other material?**

  In using the term "working memory load", we were thinking specifically of the integration and storage cost of new discourse referents, and of interference. While high working memory load also causes decay, we have assumed that the effect of decay investigated in our experiments stems primarily from time (since the distance-inducing interveners don't contain new discourse referents or interference).

## Introduction

- **I think the presentation of surprisal and related ideas is still somewhat confusing, mainly because, I believe, there are two distinct, important predictions made by surprisal, but the presentation sort of mixes the two: 1. First, surprisal says "more predictable is easier". This is by no means something that was first claimed by surprisal theory. It was shown in ERP studies since Kutas & Hilliard (1980) and in reading times since Ehrlich & Rayner (1981) (and maybe before?).**

Surprisal is just one way to model this observation. This generalization is the basic tenet in the predictions of both theories contrasted in Figure 1 (in both, reading times on the left are shorter than on the right). So I think in the subsection discussing word predictability, the discussion shouldn't really start with or focus exclusively on surprisal. In fact, on the next page the authors offer an explanation for the effect of predictability based on decay (lines 116-118). The discussion in the subsection discussing word predictability can therefore outline the main observation (predictable is easier) and findings, and then mention the surprisal account, and also the decay account. 2. The second thing, which is more specific to surprisal, is the prediction for antilocality effects. Antilocality is briefly explained in the abstract and then is sort of assumed, but never really presented methodically. So I would suggest including an explanation of this effect in a dedicated subsection. So the order will be: the "word predictability" section (the prediction of which are identical for the two hypotheses); the "antilocality" section (suprisal); and then the "decay" section.

We have rearranged the introduction into three parts following the reviewer's suggestion: 1) We introduce the idea of preactivation and how it relates to predictability, and discuss evidence for lexical preactivation in long-distance dependency formation. 2) Here we introduce the idea of adding distance within the dependency and explicitly define locality and antilocality and their relevant theoretical accounts, including suprisal. 3) Here we introduce decay as a subsection of 2). Since these changes are fairly substantial, we do not provide excerpts below, but instead refer reviewers back to the main document (or tracked changes document).

- **Also, the two paragraphs on the interaction of predictability with distance (p. 2 line 65 onward) are very confusing. They sort of go back and forth between discussing the interaction of predictability with distance and discussing the interaction of predictability with working memory load without explicitly explaining why or whether the two (distance/working memory load) are interchangeable.**

We have revised these paragraphs to make the wording more consistent on page 3; i.e. we use the term working memory load instead of distance as much as possible:

"The sources underlying antilocality and locality effects – predictability and working memory load respectively – may even interact. There is some evidence that the negative effect of high working memory load may only be apparent in weakly predictive contexts and that otherwise, antilocality effects are observed (Husain et al., 2014; Konieczny, 2000; Levy and Keller, 2013). For example, in German, it was found that reading times at the clause-final verb of a relative clause were faster when the verb was delayed by one additional constituent than when it was not delayed (an antilocality effect), but that reading times slowed down when the verb was delayed by two additional constituents (a locality effect; Levy and Keller, 2013). The authors reasoned that the relative infrequency of adding the second constituent (according to a corpus analysis) actually reduced predictability, making the effects of increased working memory load more pronounced. Casting doubt on these results, however, is a replication attempt finding only locality effects, regardless of what information preceded the verb (Vasishth et al., 2018).

More direct tests of an interaction between predictability and working memory load have been conducted in Hindi and Persian. In Hindi, increasing the separation within noun-verb complex predicate facilitated the reading of highly predictable verbs, but slowed the reading of low-predictable verbs, suggesting that high predictability outweighed the effect of additional working memory load introduced by the intervening sentence material (Husain et al., 2014). However, this load/predictability interaction was not replicated in analogous constructions in Persian, where higher working memory load induced by additional sentence material slowed reading of the distant verb, regardless of the verb's predictability (Safavi et al., 2016). One difference between the Hindi and Persian studies was the type of information used to manipulate the separation distance of the complex predicate dependencies. The Persian study used a relative clause and a prepositional phrase as an intervener (Safavi et al., 2016). Both relative clauses and prepositional phrases introduce new discourse referents and interference, both of which are predicted to burden

working memory resources and slow reading (Gibson, 1998, 2000; Lewis and Vasishth, 2005), although new discourse referents may not be the only source of slowing in longer dependencies (Gibson and Wu, 2013). In comparison, the separation in the Hindi experiments was increased with adverbials, which instead may have increased evidence for the position and lexical identity of the upcoming verb (Hale, 2001; Levy, 2008). Altogether, these findings suggest that while readers may preactivate the lexical entry of an upcoming dependent word, if appearance of that word is delayed, its predictability may play an important role in how the intervening information impacts processing."

- **Finally, the bottom line of these two paragraphs is "facilitation in the reading times of a distant word .. may only occur when that word is highly predictable" (this is also stated in the predictions section, and in the conclusion) – but this interaction is not represented in Figure 1, where the effect of distance is identical for more predictable and less predictable words.**

The surprisal predictions in Figure 1 are intended to reflect the canonical surprisal account, where long distance = faster reading time, regardless of working memory load or some other factor. We intended for the two paragraphs above to describe situations where surprisal's prediction about distance might be too simplistic (i.e. under varying working memory load), even though we don't actually use a working memory load manipulation in our experiments. Thus, for our experiments, the predictions of canonical surprisal as presented in Figure 1 should still stand. The point of including these paragraphs at all was to underscore why it was important that we used interveners that only extended linear distance between the verb and particle, without providing additional clues about the particle's identity or adding extra working memory load (insofar as that is possible). We hope that the revised Introduction makes this clearer.

- **One general suggestion: it would perhaps be helpful to have one example sentence in the introduction (perhaps even with a verb-particle dependency) to accompany the discussion, so the different predictions can be exemplified with regard to that sentence, to make them concrete and easier to under-**

**stand.**

The revision of the Introduction now makes our predictions more explicit. For example, to sum up the paragraph on antilocality on page 3, we now explicity state that:

> "Thus, surprisal predicts that the longer the distance separating two dependent words, the more expected and easy to process the distant word will become."

Then, in the section on decay, as per the Reviewer's suggestion, we have added a particle verb example to illustrate exactly how predictability could interact with decay:

> "The above example concerns plausible structural continuations of the sentence, but plausible continuations may also include the preactivation of specific lexical items. For example, in 1a below, the verb *turn* may trigger preactivation of plausible sentence continuations, including a large number of frequent particles (turn off, turn on, turn around, turn over, etc.). If the sentence continues with *the music*, preactivation should be constrained to a smaller group of plausible particles:

> (1)  a.  Turn the music... [on, off, up, down]
>      b.  Calm the situation... [down]

> A specific particle may even be pre-integrated while the others are left to decay. If future input indicates that the wrong particle was pre-integrated, e.g. *up* instead of *down*, then *down* must be reactivated in order to repair the sentence, resulting in longer reading times at the particle. As the number of plausible lexical items increases, reading times should therefore become slower on average, because the probability that the parser pursues a parse with the wrong lexical item increases and reactivation of decayed items will be needed more often. Alternatively, the starting activation of *down* in 1a may be lower than that of *down* in 1b, because the latter context points strongly to *down* as the only plausible continuation. The stronger starting activation of *down* in 1b should mean that even as activation decays over time, it will still have stronger activation at matched points in the sentence than in 1a. Thus, overall, more predictable lexical items should be more resistant to the effects of decay than less predictable items."

## Experiment 1 Methods

- **I think it would be helpful to state explicitly, around line 307, that the set size manipulation therefore did not result in a difference in the predictability of the particle.**

  An explicit statement has now been included in the following paragraph on page 9:

  > "This analysis raised an immediate problem with the experimental design. The categorical predictor *set size* used in the planned analysis was intended as a proxy for entropy and predictability, where a large set size was supposed to reflect high entropy and thus lower predictability. However, although these categories may have reflected the number of particles licensed by each base verb, the results of the cloze test suggested they did not represent the range of particle completions provided by readers at the particle site. This can be seen in Figure 3: although the *average* entropy was higher in the large set than in the small set condition, both conditions contained high and low entropy sentences. In other words, there was no difference in predictability of the particle between the small and large set conditions."

- **The entropy formula on line 320 should be explained.**

  We have now moved the entropy formula to a footnote, with the following explanation on page 8:

  > "Entropy (H) was calculated as the negative sum of cloze probabilities (P) for all particles provided by participants for a particular sentence in the cloze test, multiplied by their respective logs: $H = -\sum_i P_i log_2 P_i$. For example, if nine cloze completions were the particle "vor" and one was "an", then: $H = -(P_{vor} \cdot log_2 P_{vor} + P_{an} \cdot log_2 P_{an}) = -(0.9 \cdot log_2 0.9 + 0.1 \cdot log_2 0.1) = 0.47$"

- **One last thing – just a thought, no need to do this – I wonder whether in the eyetracking there would be effects on rates of skipping the particle altogether (since we know that more predictable words are skipped more often). The authors say that the particle was not always fixated – I wonder, for future studies, if there could be something interesting there.**

We did actually look at this and, interestingly, although the particle was not always fixated, it was fixated more often than we anticipated. Skipping rates were therefore correspondingly low. We believe this may have to do with the particle appearing at the right clause boundary adjacent to a comma. Scanpath analysis might be an interesting way to look at this in the future.

## Typographical errors and very minor comments

The following typographical errors and suggestions have been amended:

- **Abstract, 8th line from bottom: should be "decay, predictability or their interaction".**

- **Abstract, 5th line from bottom: perhaps instead of "facilitate or hinder reading times", change to "facilitate or hinder processing"?**

- **Line 210: parentheses missing around Lewis and Vasishth, 2005.**

# References

Gibson, E. (1998). Linguistic complexity: Locality of syntactic dependencies. *Cognition*, 68(1):1–76.

Gibson, E. (2000). The Dependency Locality Theory : A Distance -Based Theory of Linguistic Complexity. In Marantz, A., Miyashita, Y., and O'Neil, W., editors, *Image, Language, Brain*, pages 95–126. MIT Press.

Gibson, E. and Wu, H.-H. I. (2013). Processing Chinese relative clauses in context. *Language and Cognitive Processes*, 28(1-2):125–155.

Hale, J. (2001). A probabilistic Earley parser as a psycholinguistic model. *NAACL '01: Second meeting of the North American Chapter of the Association for Computational Linguistics on Language technologies 2001*, pages 1–8.

Husain, S., Vasishth, S., and Srinivasan, N. (2014). Strong expectations cancel locality effects: Evidence from Hindi. *PloS one*, 9(7):e100986.

Konieczny, L. (2000). Locality and parsing complexity. *Journal of Psycholinguistic Research*, 29(6):627–45.

Levy, R. (2008). Expectation-based syntactic comprehension. *Cognition*, 106(3):1126–1177.

Levy, R. and Keller, F. (2013). Expectation and locality effects in German verb-final structures. *Journal of Memory and Language*, 68(2):199–222.

Lewis, R. L. and Vasishth, S. (2005). An activation-based model of sentence processing as skilled memory retrieval. *Cognitive science*, 29(3):375–419.

Safavi, M. S., Husain, S., and Vasishth, S. (2016). Dependency resolution difficulty increases with distance in Persian separable complex predicates : Evidence against the expectation-based account. *Frontiers in Psychology*, pages 1–21.

Vasishth, S., Mertzen, D., Jäger, L. A., and Gelman, A. (2018). The statistical significance filter leads to overoptimistic expectations of replicability. *Journal of Memory and Language*, 103:151–175.

---

## Round 0.4 · accepted · Accept

Dear Kate and fellow authors,

Thank you for your latest round of revisions. I have looked at your rebuttal, as well as your marked up PDF, and all looks good to me. If I have missed any minor issues, I am sure you and the typesetters will pick them up in the next phases of processing prior to publication.

Congratulations on all your hard work. I look forward to seeing this in print. Oh, and it is great to know that now at least two people in the world (ie you and I) will now giggle whenever they read details about posterior distributions. We all need a laugh right now.

With the best of wishes to you and your team,

Genevieve

---

## Author Rebuttal · Round 0.4

Department of Linguistics
University of Potsdam
Karl-Liebknecht-Straße 24-25
14476 Potsdam, Germany

Prof Genevieve McArthur
Associate Editor
Dear Prof McArthur,

Thank you for your response regarding our manuscript and we are very pleased to hear that both you and Reviewer 2 were satisfied with the last revision.

We have addressed your suggestions as described point by point below. We're very grateful for your feedback and have found the review process very helpful and constructive, thank you!

Yours sincerely,

Dr. Kate Stone
Dr. Titus von der Malsburg
Prof. Shravan Vasishth

# Responses to comments

- **There are a few sentences that require some minor adjustments in wording to fix the grammar. I have highlighted these sentences in the attached PDF.**

  These have been amended, thank you for the suggestions. Two exceptions were:

  - In the conclusions, it was suggested to change "...evidence against an effect" to "no evidence for an effect". This wording relates to the type of statistical inference criterion we have used (the Bayes factor), which actually allows us to make statements about evidence against the null because it directly quantifies evidence for/against the null and alternative hypotheses.
  - The use of "processing "at" a word", rather than "of": Here we would prefer to retain the use of "at" to reflect that processing difficulty is measured at a specific word, but does not necessarily entail that the effect seen is processing "of" that specific word (e.g. it could be a processing spillover from a previous word, even though we're basing inference on effects seen at a target word).

- **Throughout the manuscript, there is inconsistent use of hyphen between particle and verb (e.g., particle verb and particle-verb and vice versa). Choose one convention and use consistently.**

  We have made this as consistent as possible, however there are some occasions when it is more appropriate to talk about the name of the construction (particle verb), and others where it is more appropriate to talk about the dependency between the verb and particle (verb-particle dependency). We have therefore revised the text to be consistent between these two cases.

- **I am pretty sure PeerJ will want references in parentheses in alphabetical order. Please go through and amend throughout the manuscript. Also, please double-check the formatting for references in the text, and make sure you use PeerJ conventions in terms of the use of "and" or "&" etc.**

  [** PeerJ Staff Note - as long as the references are complete, the formatting will be done during typesetting **]

Since the referencing and citations are auto-formatted by the PeerJ Latex style template, we are hesitant to edit this template file. In line with the PeerJ staff note, we will provide a full bibtex file so that the references can be typeset according to PeerJ policy.

- **The use of the term "surprisal" is grammatically incorrect in English. Surprisal account is perfectly OK. I suggest you identify all sentences that include the word "surprisal" on its own, and revise them to accommodate "surprisal account" in a grammatically appropriate way.**

We have now revised the text to explicitly state "surprisal theory" or "surprisal account" where relevant. In some specific contexts, there is a technical difference between "surprisal theory" and a word's surprisal value. In the latter case, it is necessary to use the word "surprisal" alone in order to be consistent with the literature on surprisal theory. However, this distinction should be clear now that all other cases use "surprisal theory".

- **Your manuscript is very "dense" in terms of terminology. Please avoid the use of acronyms in the text (e.g., NP, BF) because it makes it just that much harder for the reader to follow your meaning.**

We have now spelled out all acronyms or, in one case, provided definitions in the figure caption where this was not possible (see below).

- **I am not sure why the distributions throughout the manuscript are called "posteriors". I suggest you just called them distributions (google posteriors and you will see why - actually, don't!! do that - just look up the meaning of posteriors).**

The term "posterior distribution" is a technical term from Bayesian statistics. The term "posterior distribution" refers to its temporal relationship with the "prior distribution", both of which are involved in a Bayesian analysis. The prior is a distribution that must be specified before seeing the data and the posterior is what results after seeing the data. For that reason, it's unfortunately not possible to call the posterior anything else - but now that I have the association with the other meaning of posterior, I will certainly giggle every time I have to use it!

- **Table 8 and similar. Please provide full version of acronyms**

**either in table (you may have room in column 1) or in the
title or the notes.**

Acronyms for the eye tracking measures have now been spelled out
fully everywhere except Figure 8, which had space limitations. For
this figure, we now spell out the acronyms in the figure caption.

- **When putting things in lists, make sure you used the PeerJ
  formatting for numbers (e.g., i, ii, iii OR (1), (2), (3) etc).**

  The in-text lists have now been made consistent as (i), (ii), etc. Brack-
  ets were used to distinguish the list items from the text. The linguistic
  examples use the numbering common in linguistics, e.g. in line with
  the gb4e Latex package (Kolb & Thiersch, 2010).

- **If a number is less than 10, then write in full (e.g., nine).**

  This has been amended throughout.

## Additional changes

- **One substantive change to the text:**

  We have deleted the following sentence that began on line 118 of the
  previous manuscript version:

  "Lexical constraints are often not explicitly modelled in surprisal (Levy,
  2008; Hale, 2001), but lexicalised PCFGs have demonstrated that the
  contribution of lexical information to processing difficulty follows a
  similar pattern to the canonical syntactic model (Collins, 2003; Char-
  niak, 2001)."

  The reason for deleting this sentence was that, in a previous manuscript
  version, we had begun by talking about syntactic surprisal and so it
  was necessary to state that lexical surprisal functions in a similar way.
  However, the preceding sentence has now been simplified such that it
  covers both syntactic and lexical surprisal, and so the above sentence
  is no longer necessary (and perhaps even adds confusion).